

# The Holocene dynamics of Ryder Glacier and ice tongue in north Greenland

Matt O'Regan[1,2*], Thomas M. Cronin[3], Brendan Reilly[4], Aage Kristian Olsen Alstrup[5], Laura Gemery[3], Anna Golub[3], Larry A. Mayer[6], Mathieu Morlighem[7], Matthias Moros[8], Ole L. Munk[5], Johan Nilsson[2,9], Christof Pearce[10], Henrieka Detlef[10], Christian Stranne[1,2], Flor Vermassen[1,2], Gabriel West[1,2], and Martin Jakobsson[1,2]

[1]Department of Geological Sciences, Stockholm University, 10691, Stockholm, Sweden
[2]Bolin Centre for Climate Research, Stockholm University, 10691, Stockholm, Sweden
[3]Florence Bascom Geoscience Center, U.S. Geological Survey, Reston, VA, 20192, USA
[4]Scripps Institution of Oceanography, University of California San Diego, La Jolla, CA, 92037, USA
[5]Department of Clinical Medicine - Nuclear Medicine and PET, Aarhus University
[6]Center for Coastal and Ocean Mapping, University of New Hampshire, Durham, NH, 03824, USA
[7]Department of Earth System Science, University of California, Irvine, CA, 92697, USA
[8]Leibniz Institute for Baltic Sea Research Warnemünde, D-18119, Rostock, Germany
[9]Department of Meteorology, Stockholm University, 10691, Stockholm, Sweden
[10]Department of Geoscience and Arctic Research Centre, Aarhus University, 8000, Aarhus, Denmark

*Correspondence to*: Matt O'Regan (matt.oregan@geo.su.se)

**Abstract.** The northern sector of the Greenland ice sheet is considered to be particularly susceptible to ice mass loss arising from increased glacier discharge in the coming decades. However, the past extent and dynamics of outlet glaciers in this region, and hence their vulnerability to climate change, are poorly documented. In the summer of 2019, the Swedish icebreaker *Oden* entered the previously unchartered waters of Sherard Osborn Fjord, where Ryder Glacier drains approximately 2% of Greenland's ice sheet into the Lincoln Sea. Here we reconstruct the Holocene dynamics of Ryder Glacier and its ice tongue by combining radiocarbon dating with sedimentary facies analyses along a 45 km transect of marine sediment cores collected between the modern ice tongue margin and the mouth of the fjord. The results illustrate that Ryder Glacier retreated from a grounded position at the fjord mouth during the Early Holocene (>10.7 ± 0.4 cal ka BP) and receded more than 120 km to the end of Sherard Osborn Fjord by the Middle Holocene (6.3 ± 0.3 cal ka BP), likely becoming completely land-based. A re-advance of Ryder Glacier occurred in the Late Holocene, becoming marine-based around 3.9 ± 0.4 cal ka BP. An ice tongue, similar in extent to its current position was established in the Late Holocene (between 3.6 ± 0.4 and 2.9 ± 0.4 cal ka BP) and extended to its maximum historical position near the fjord mouth around 0.9 ± 0.3 cal ka BP. Laminated, clast-poor sediments were deposited during the entire retreat and regrowth phases, suggesting the persistence of an ice tongue that only collapsed when the glacier retreated behind a prominent topographic high at the landward end of the fjord. Sherard Osborn Fjord narrows inland, is constrained by steep-sided cliffs, contains a number of bathymetric pinning points that also shield the modern ice tongue and grounding zone from warm Atlantic waters, and has a shallowing inland sub-ice topography. These features are conducive to glacier stability and can explain the persistence of





Ryder's ice tongue while the glacier remained marine-based. However, the physiography of the fjord did not halt the dramatic retreat of Ryder Glacier under the relatively mild changes in climate forcing during the Holocene. Presently, Ryder Glacier is grounded more than 40 km seaward of its inferred position during the Middle Holocene, highlighting the potential for substantial retreat in response to ongoing climate change.

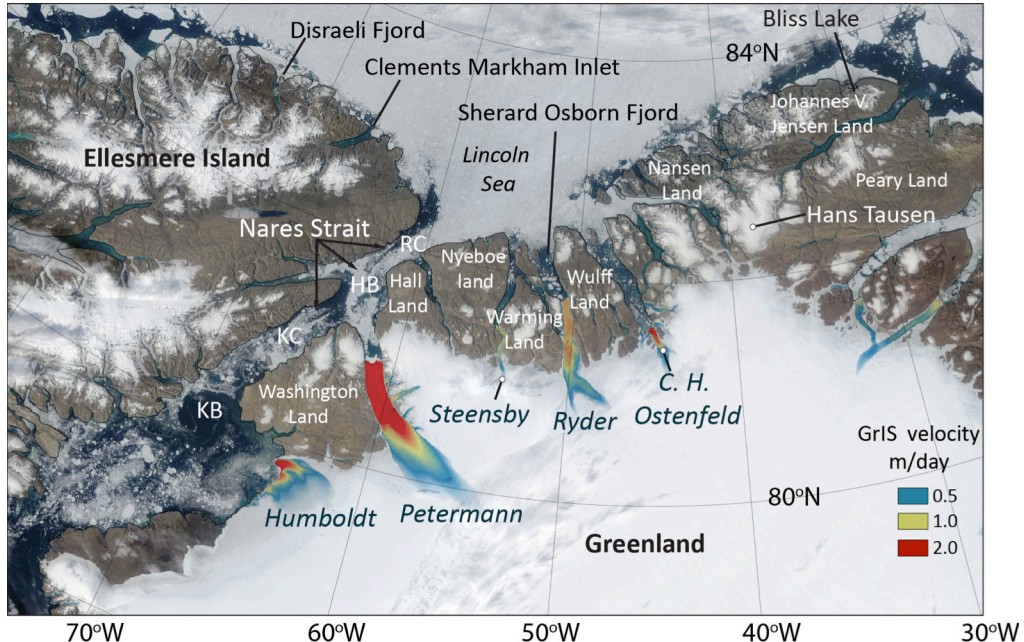

**Figure 1: Satellite view of north Greenland and Ellesmere Island from 07/30/2019. Velocity of outlet glaciers are from Sentinel-1, winter campaign 2019/2020 [version 1.3] (Nagler et al., 2015). Robeson Channel (RC), Hall Basin (HB), Kennedy Channel (KC), Kane Basin (KB), Greenland Ice Sheet (GrIS).**

## 1. Introduction

Mass loss from the Greenland ice sheet (GrIS) occurs from surface ablation (melting) and through iceberg calving (discharge) and subaqueous melt at marine terminating glaciers. It has increased six-fold since the 1980's, contributing an estimated 13.7 mm to global sea level between 1972-2019 (Mouginot et al., 2019). In north and northeast Greenland (Fig. 1), ice discharge rates from marine terminating glaciers are lower than those observed in the south and northwest (Mouginot et al., 2019). With amplified rates of high Arctic warming, and the continued loss of sea ice and buttressing ice shelves, accelerated ice discharge from the northern sector of the GrIS has been identified as a particular concern for sea-level rise in the coming decades (Moon et al., 2012; Hill et al., 2018; Mouginot et al., 2019).



Marine sediment archives provide unique insights into the past extent and dynamics of Greenland's marine-based ice margin, including the sensitivity and environmental controls on the stability of outlet glaciers and their floating ice tongues (Jakobsson et al., 2018; 2020; Wangner et al., 2018; Reilly et al., 2019; Vermassen et al., 2020). Combined with analyses of ice cores (Vinther et al., 2009, Lecavalier et al., 2017), terrestrial and marine mapping of glacial limits (Funder et al., 2011a),

and other paleoclimate time-series (Briner et al., 2016), marine sediment archives allow us to investigate the response of the GrIS to natural climate variability over time scales that exceed the length of direct satellite observations and other historical records. An understanding of how marine terminating glaciers responded to past climate change, and ultimately elucidating the geologic and environmental controls on their behaviour, are critical to reduce uncertainties in future sea-level predictions (Bamber et al., 2019).


After the last glacial maximum, the GrIS receded dramatically through the Early (11.7 – 8.2 cal ka BP) and Middle Holocene (8.2-4.2 cal ka BP) when Arctic summer air temperatures were ~1-3 °C above 20th century averages (Kaufman et al., 2004; Miller et al., 2010; Briner et al., 2016). The GrIS reached its minimum extent near the end of the Middle Holocene (8.2-4.2 cal ka BP) or sometime near the beginning of the Late Holocene (4.2-0 cal ka BP) (Young and Briner 2015). The Lincoln

Sea, which surrounds much of northern Greenland hosts some of the most persistent and harsh sea-ice conditions in the Arctic today. However, many terrestrial glacial (Kelly and Bennike, 1992; Landvik et al., 2001; Molner et al., 2010; Funder et al, 2011b; Larsen et al., 2019) and lacustrine paleoclimate studies (Olsen et al., 2012) show warmer climatic conditions, reduced glacial ice extent, and more open water conditions along the north Greenland coast during the Holocene Thermal Maximum (11.0-5.5 cal ka BP). Due to difficulties in accessing the Lincoln Sea, there are no marine records documenting

glacier dynamics north of Petermann Glacier (Reilly et al., 2019) that can be combined with these land-based studies.

In the summer of 2019, during the *Ryder 2019* expedition, the Swedish icebreaker *Oden* became the first vessel to enter the unchartered waters of Sherard Osborn Fjord, which connects Ryder Glacier with the Lincoln Sea (Fig. 1). Ryder Glacier drains about 2% of Greenland's ice sheet and is one of four major marine terminating glaciers in this sector of the GrIS (Fig.

1). Ryder and Petermann in the northwest and 79°-Glacier in the northeast are the only remaining Greenland outlet glaciers that have large, intact floating ice tongues, believed to exert an important buttressing force that slows glacier flow (Mottram et al., 2019). Ryder's ice tongue is 25 kilometers long and has been relatively stable during the last 70 years, showing a net advance of about 43 m a$^{-1}$ between 1948 and 2015 (Hill et al., 2018).

Here we combine radiocarbon dating with the analysis of lithofacies in six marine sediment cores that form a ~45 km long transect extending from the modern ice tongue margin of Ryder Glacier to the mouth of Sherard Osborn Fjord. We integrate these results with established phases of ice recession and re-growth in this sector of northern Greenland, providing the first insights into the dynamic behavior of Ryder Glacier and its ice tongue during the Holocene.





## 2. Geologic, oceanographic and glaciologic setting

Sherard Osborn Fjord is ~17 km wide and extends ~55 km from the ice tongue margin of Ryder Glacier out towards the Lincoln Sea. Ryder Glacier is currently grounded below sea level, with the grounding zone located ~26 km landward of the ice tongue terminus (Fig. 2). Bathymetric mapping during the *Ryder 2019* expedition revealed two prominent sills dissecting the fjord (Jakobsson et al., 2020) (Fig.2). These sills bound an over-deepened inner basin that has a maximum depth of 890 m. The outer sill has little sedimentary cover and appears to be a bedrock feature, while the inner sill is interpreted as a sedimented former glacial grounding zone (Jakobsson et al., 2020), potentially developed on a pre-existing bedrock high.

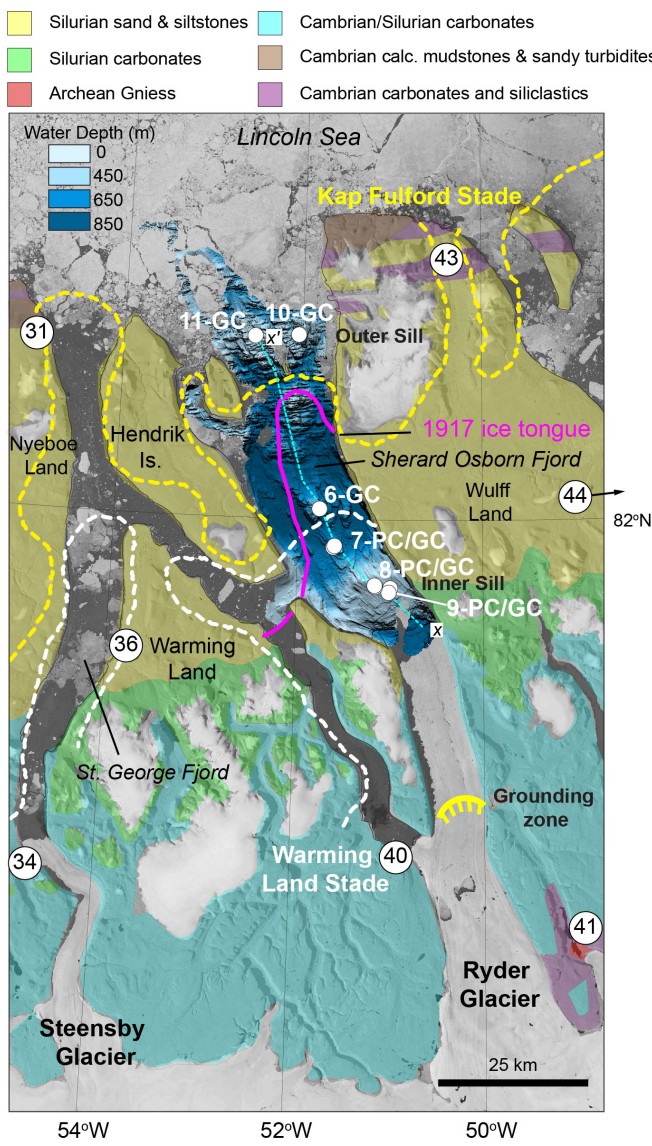

**Figure 2. Landsat image of Sherard Osborn Fjord from 30/07/2019 showing locations of cores discussed in this study. Bedrock geology is from Henriksen et al. (2009). Bathymetry of Sherard Osborn Fjord was collected during the *Ryder 2019* expedition (Jakobsson et al., 2020). Ice limits for the Kap Fulford (yellow) and Warming Land (white) Stades are redrawn from Kelly and Bennike (1992). The extent of Ryder ice tongue in 1917 (pink) is re-drawn from Koch (1928). Numbers refer to key locations of radiocarbon dates used by Kelly and Bennike (1992) to constrain Holocene ice margin positions (Table 1). Dashed cyan line (x-x') marks the location of the oceanographic profile in Fig. 3.**





The sill depths on the outer fjord are 475 m on the east and 375 m on the west (Fig. 2). The inner sill has a ~6.2 km wide central region that ranges in depth from 193 to 300 m, with a ~1 km wide channel on the eastern side that extends to a depth

of 390 m. The modern ice tongue terminus of Ryder Glacier is located approximately 5 km landward of the inner sill. Despite these bathymetric barriers, waters of Atlantic origin that circulate through the Lincoln Sea are found between the two sills at depths greater than 350 m (Fig. 3; Jakobsson et al., 2020). These relatively warm (>0.3°C) and saline (>34.7) waters are constrained by the inner sill with only a small amount of warmer water flowing across the inner sill through the <1 km wide and 390 m deep channel, to be strongly mixed with glacially derived meltwater.


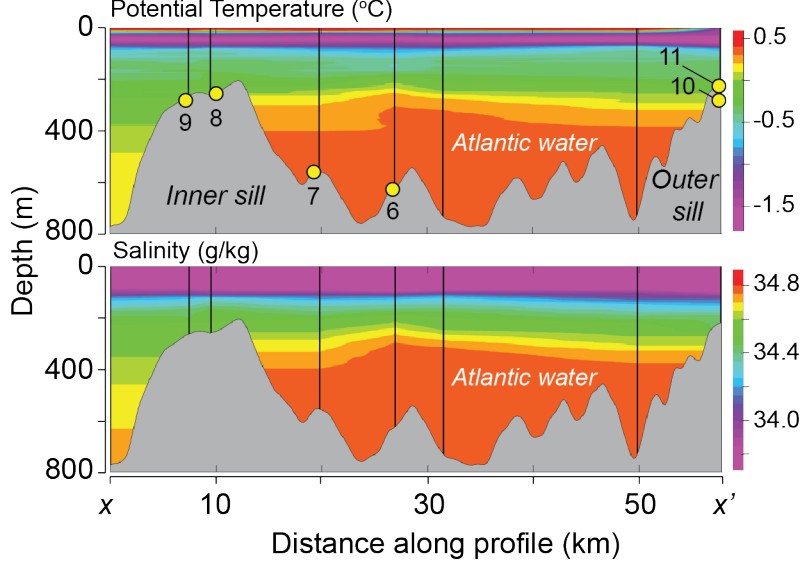

Figure 3. Potential temperature and salinity profiles illustrating the distribution of Atlantic water in Sherard Osborn Fjord. Vertical black lines are positions of CTD casts conducted during *Ryder 19*. Yellow circles are the locations of coring sites. The position of site 10 on the outer sill is projected, as this profile does not cross its location (see supplementary for additional details).

Sherard Osborn Fjord cuts into the Paleozoic Franklinian Basin that extends over 2000 km from the Canadian Arctic to eastern north Greenland (Henriksen and Higgins, 2000). Along the north Greenland coast, deposition in the Franklinian Basin occurred in a passive margin environment through the latest Precambrian to Devonian, with shallow water carbonate

shelf sediments found to the south and siliciclastic slope and deep-water sediments in the north (Henriksen et al., 2009). Exposed sedimentary bedrock between the present GrIS margin and the terminus of the floating ice tongue of Ryder Glacier is composed of Cambrian and Silurian carbonate shelf sediments. Further seaward, Silurian sands and siltstones deposited in a deeper water turbiditic environment are found on the lowlands of Nyeboe and Wulff Land (Henriksen and Higgins, 2000). Older, deformed Cambrian deep-water calcareous mudstones and sandy turbidites cap the outer headlands of Nyeboe and

Wulff Land, and are part of the 600 km long E-W trending North Greenland fold belt that formed near the end-Devonian Ellesmerian Orogeny (Higgins et al., 1998) (Fig. 2).



Terrestrial mapping of glacial landforms and marine limits across Nyeboe Land, Warming Land and Wulff Land has defined three regional glacial events from the end of the Pleistocene (Late Weichselian) through the Holocene (Kelly and Bennike,

1992). The oldest marginal ice limits that are traced across north Greenland belong to the Kap Fulford Stade (Kelly and Bennike, 1992). During this time, the GrIS extended across the southern parts of many peninsulas, with outlet glaciers flowing to the outer limits of many of the fjords. Radiocarbon dates from marine macrofossils sampled by Kelly and Bennike at Kap Fulford (Station 31; 10030 ± 175 and 9390 ± 90 $^{14}$C a BP) and freshwater algae from northern Wulff Land (Station 43; 10 480 ± 90 $^{14}$C a BP) provide minimum age estimates for the retreat of glacial ice of the Kap Fulford Stade to the latest

Pleistocene to Early Holocene (>10.5 cal ka BP) (Kelly and Bennike, 1992) (Table 1, Fig. 2).

**Table 1. Key radiocarbon dates originally used to constrain the ages of glacial stages in the vicinity of Sherard Osborn Fjord. Index numbers refer to original site numbers given in Kelly and Bennike (1992) with locations shown on Figure 2.**

| # | Area | Relevance | Elev. (m) | Material | Lab. ID | $^{14}$C age (a) |
|---|------|-----------|-----------|----------|---------|------------------|
| 31 | Nyeboe land | Kap Fulford Stade | 87 | shell | K-4339 | 10030 ± 175 |
| 34 | Nyeboe land | Steensby Stage | 24 | shell | K-4380 | 4870 ± 80 |
| 36 | Warming Land | Warming Land Stade | 69 | shell | HAR-6290 | 8210 ±120 |
| 40 | Warming Land | Warming Land Stade | 26 | shell | HAR-6287 | 6480 ± 100 |
| 41 | Wulff Land | Steensby Stade | 275 | peat | K-4573 | 5100 ± 130 |
| 43 | Wulff Land | Kap Fulford Stade | 72 | algae | GU-2588 | 10480 ± 90 |
| 44 | Wulff Land | Warming Land Stade | 62 | shell | K-4374 | 8000 ± 115 |

A second set of ice marginal deposits are found along many of the fjords, 20-60 km inland from the Kap Fulford Stade. These are assigned to the Warming Land Stade, representing a standstill of outlet glaciers draining the retreating ice front. Regionally, the age of the Warming Land Stade is bracketed between >9.5 cal ka BP to 8.0 cal ka BP (Kelly and Bennike, 1992). Many glacially dammed lakes formed during the Warming Land Stade, as glaciers continued to occupy the fjord systems while the ice margin retreated on land (Kelly and Bennike, 1992). For example, an ice-dammed lake formed on

Wulff Land, in the central lowlands between Ryder and C. H. Ostenfeld glaciers (Fig. 1). The most proximal minimum age constraints for the Warming Land Stade in Sherard Osborn Fjord come from southeast Warming Land, where marine macrofossils in sediments younger than the ice margin deposits provide an age of 6480±100 $^{14}$C a BP (Station 40, Kelly and Bennike, 1992). On the eastern edge of Wulff Land outlet glaciers built a series of deltas in the ice-dammed lake during the Warming Land Stade. Marine macrofossils in the lowest delta are dated to 8000±115 $^{14}$C a BP (Station 44) indicating that

the retreat of C. H. Ostenfeld Glacier and drainage of the glacial lake occurred after this time (Kelly and Bennike, 1992). The ice margin continued to retreat following the Warming Land Stade. In the vicinity of Ryder glacier, it likely reached a position equivalent to its modern one by the Middle Holocene ~ 6 cal ka BP, before receding even further inland (Kelly and Bennike, 1992).



The Steensby Stade marks the most recent re-advance of the ice margin, outlet glaciers and local ice caps to their maximum positions since the Kap Fulford Stade. Its onset is poorly dated and occurred sometime during neoglacial cooling that extent following the Holocene Thermal Maximum. At the GrIS margin to the north of Ryder Glacier, peat deposits over which the ice margin advanced provide an age of 5100±130 [14]C a BP (Station 41), while at Steensby Glacier, reworked marine macrofossils in lateral moraines yield an age of 4870±80 [14]C a BP (Station 34; Kelly and Bennike, 1992). Despite

uncertainty surrounding the onset of the Steensby Stade, and subsequent dynamics of Ryder Glacier through the Late Holocene, the maximum extent is believed to coincide with historical observations that place the terminus of outlet glaciers and ice tongues near the mouth of many of the major fjords. This is true for Ryder's ice tongue, which was positioned near the outer margin of Sherard Osborn Fjord in 1917 by the Danish geologist and explorer Lauge Koch (Koch, 1928) (Fig. 2). Between 1917 and 1947, Ryder ice tongue retreated to near its current position (Davies and Krinsley, 1962; Higgins, 1990),

and has remained relatively stable, even exerting a net advance of 43 m a$^{-1}$ between 1948-2015 (Hill et al., 2018).

## 3. Materials and methods

### 3.1 Marine sediment cores

Sediment cores from Sherard Osborn Fjord were collected during the 37-day *Ryder 2019* expedition (August 5-September 10, 2019). Between August 13 and 25, unusually light sea-ice conditions allowed *Oden* to occupy nine coring stations and

systematically map the fjord (Fig. 2; Jakobsson et al., 2020). Coring was conducted using a 12 m long piston core (PC) and small (1-2 m) trigger weight core (TWC), and a 6 m long gravity core (GC). Piston and gravity cores were collected in liners with an inner/outer diameter of 100/110 mm, while TWCs were collected in narrower (80/88mm) liners.

**Table 2. Locations, water depths and lengths of cores used in this study.**

| Station | Core ID | Latitude (°N) | Longitude (°E) | Water Depth (m) | Core Length (m) |
|---|---|---|---|---|---|
| **6** | Ryder19-6-GC1 | 82.0095 | -51.7408 | 633 | 4.93 |
| **7** | Ryder19-7-GC1 | 81.9532 | -51.5760 | 551 | 5.19 |
| | Ryder19-7-PC1 | 81.9518 | -51.5878 | 559 | 8.96 |
| | Ryder19-7-TWC1 | 81.9518 | -51.5878 | 559 | 0.96 |
| **8** | Ryder19-8-GC1 | 81.8947 | -51.1365 | 228 | 4.98 |
| | Ryder19-8-PC1 | 81.8928 | -51.1315 | 238 | 8.97 |
| **9** | Ryder19-9-GC1 | 81.8843 | -50.9848 | 271 | 5.89 |
| | Ryder19-9-PC1 | 81.8908 | -50.9682 | 274 | 8.72 |
| **10** | Ryder19-10-GC1 | 82.2713 | -52.0165 | 272 | 2.87 |
| **11** | Ryder19-11-GC1 | 82.2682 | -52.5038 | 208 | 1.34 |




Here we focus on cores obtained from five stations that were positioned on topographic highs, with a two radiocarbon dates coming from a sixth station on the outer sill. The five main stations form a 45 km long transect extending from the edge of the modern ice tongue to the outer sill of Sherard Osborn Fjord (Table 2, Fig. 2). Sub-bottom profiles were acquired across

the coring sites with *Oden's* Kongsberg SBP120 (3°x3°) chirp sonar using a 2.5-7 kHz pulse. These profiles show that a relatively thin (<10-15 m) drape of sediments exists on top of the acoustic basement (Fig. 4). This basement surface could either be sedimentary bedrock, or highly lithified sedimentary units like till. Hogan et al. (2020) illustrated that in Petermann Fjord, lower frequency air gun seismic data was required to differentiate bedrock from more lithified sediments and till.

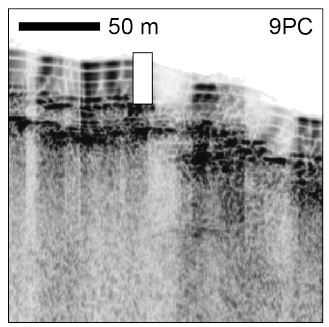
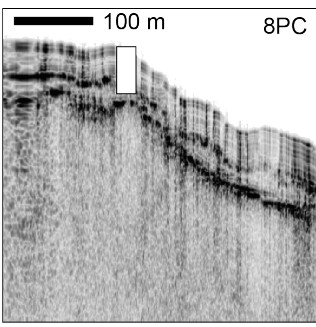
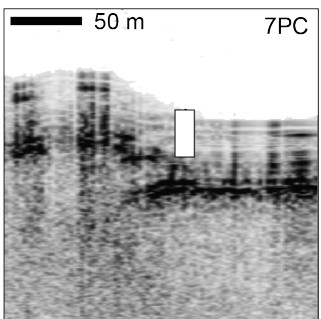

**Figure 4. Sub-bottom profiles across coring stations in Sherard Osborn Fjord illustrating the amount of the generally thin sedimentary cover on top of acoustic basement penetrated by each core.**

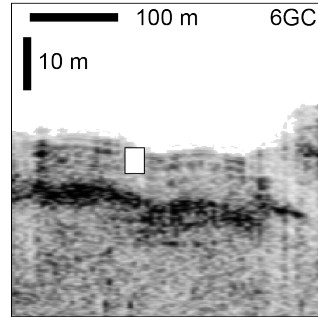
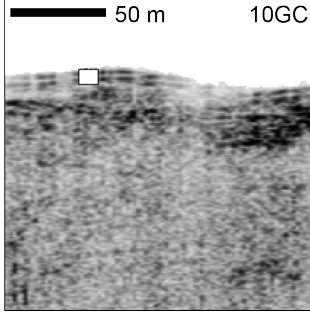


The unsplit sediment cores were allowed to equilibrate to room temperature (~20°C) and logged shipboard on a Geotek multi-sensor core logger (MSCL). The MSCL provided downcore measurements (1 cm sampling interval) of bulk density (using gamma-ray attenuation) and magnetic susceptibility. Bulk density is controlled by sediment porosity and grain composition (grain density). In predominantly lithogenic normally consolidated marine sediments, variability in bulk density

is a function of grain size-driven changes to porosity (coarser sediments = lower porosity = higher bulk density). Magnetic susceptibility is a useful proxy for the input of siliciclastic versus biogenic/organic sediments, and in dominantly lithogenic sediments can be used to discriminate between provenance and grain size (Hatfield et al., 2017; Reilly et al., 2019). Gamma-ray attenuation was measured using a [137]Cs source with a 5 mm collimator and a 10 s count time. Magnetic susceptibility was measured with a 125 mm Bartington loop sensor using a 1 s acquisition time. An empirical volume correction was



applied to account for the difference between the core diameter ($D_C$) and the loop diameter ($D_L$). The volume-specific magnetic susceptibility ($\kappa$, $10^{-5}$ SI) is defined as,

$\kappa = \kappa_{(uncorrected)}\big/\kappa_{(rel)}$

where

$\kappa_{(rel)} = 4.8566\ (D_C / D_L)^2 - 3.0163(D_C / D_L) - 0.6448$

The mass-specific magnetic susceptibility ($\chi$) was calculated by dividing $\kappa$ by the bulk density.

After logging, piston and gravity cores were split into working and archive half sections. Archive halves were described and imaged using a digital line-scanning camera on the MSCL, while the working half was sampled for shore-based analyses. Split sections were wrapped in plastic film, placed in D-tubes and stored in a refrigerated container (4°C) until they were

returned to Stockholm University's refrigerated core storage facility. TWC's remained unsplit and placed directly into the refrigerated container.

Following the expedition, all archive halves were analysed on an Itrax X-ray fluorescence (XRF) core scanner at the Department of Geological Sciences, Stockholm University. Measurements were performed with a Mo tube set to 55 kV and

50 mA with a downcore measurement resolution of 2 mm and a counting time of 15 s. Here we only present the raw counts of calcium (Ca) as a proxy for detrital input from the surrounding Cambrian and Silurian carbonate bedrock terrain.

Computed tomography (CT) scanning was performed on selected archive sections and unsplit TWC's at the Department of Nuclear Medicine & PET of Aarhus University Hospital in Denmark using a Siemens Biograph Vision 600 PET/CT. The CT

scan parameters were: 120 kV, 400 Eff mAs (no CARE Dose4D), 200 mm field of view, 0.6 mm slice thickness, filter kernel B60s (sharp). Images were processed using SedCT MATLAB tools to obtain a coronal slice through the central region of each core at an effective pixel resolution of 0.5 x 0.5 mm (Reilly et al., 2017). Clasts with >1 mm diameter were automatically counted from the 3-dimensional CT data, binned into 2 cm intervals, and normalized by core volume using the algorithm of Reilly et al. (2019) which provides an objective proxy for ice rafted debris (IRD) concentration in glaciomarine

lithofacies deposited beyond the limit of grounded ice.

Twenty-nine grain size measurements were made on core 6-GC to broadly characterize grain size spectra in the different lithologic units. Measurements were performed using a Malvern Mastersizer 3000 laser diffraction particle size analyzer at the Department of Geological Sciences, Stockholm University. Wet samples were immersed in a dispersing agent (<10%

sodium hexametaphosphate solution) and placed in an ultrasonic bath to disaggregate particles before being poured into the Malvern measurement chamber.



## 3.2 Composite depth scales

The first attempts at coring in Sherard Osborn Fjord illustrated that near surface sediments were very soft, and the top of the coring tools were sinking below the seafloor resulting in variable the recovery of near surface sediments between the gravity,
piston and trigger weight cores. At stations 7, 8 and 9, multiple coring tools were deployed, and sediments were recovered using all of these devices. A composite depth scale was developed for each station to ensure that measurements from the different cores could be integrated. The composite depth scales were developed through correlation of the MSCL bulk density, magnetic susceptibility and XRF-scanning (mainly Ca content) data and additionally through visual correlation using the CT-images (*supplementary material Table S1*).

The piston core at each station was used as an undistorted reference depth scale. Depths of the gravity and trigger weight cores were stretched, or compressed, between tie points to the piston cores. At stations 7 and 9, the piston core recovered sediments closer to the seafloor than the gravity or trigger weight cores. At station 8, correlation of MSCL and XRF data indicate that the gravity core recovered 40 cm of near-surface sediments that were either not recovered or lost from the top of
the piston core.

## 3.3 Radiocarbon dating

Radiocarbon measurements ($^{14}$C) were made on 48 samples (Table 3). Shipboard and post-cruise sampling focused on constraining the ages for lithologic units. Most radiocarbon dates were obtained on 300 specimens of the benthic foraminifera *Cassidulina neoteretis (*also called *C. teretis*, see Cronin et al. 2019)*. Sixteen dates were obtained on mixed
benthic foraminifera and one date on *Cibicides lobatulus* (Table 3). Additionally, four dates were made on specimens of the planktic foraminifera *Neogloboquadrina pachyderma* to investigate offsets between surface and bottom waters. All radiocarbon measurements were performed at the National Ocean Sciences Accelerator Mass Spectrometry (NOSAMS) facility at Woods Hole Oceanographic Institution, Massachusetts, USA. Calibration of the radiocarbon dates was performed using Oxcal v. 4.4 (Bronk Ramsey, 2009) and the Marine20 calibration curve (Heaton et al., 2020).

There is no *a priori* information on the local marine reservoir correction (ΔR) for Sherard Osborn Fjord or the Lincoln Sea. Estimates of ΔR for nearby regions vary widely and have been derived using different marine calibration curves. Results from three pre-bomb living molluscs collected in the vicinity of Thule, suggest a ΔR of 5 +/- 50 years (Mörner and Funder, 1990). Kelly and Bennike (1992) applied 150-year ΔR for marine macrofossils from the region surrounding Ryder Glacier,
as suggested for areas of northernmost Greenland by Funder et al. (1982). However, Coulthard et al. (2010) found an average ΔR of 335±85 years (using Marine09; Reimer et al., 2009) based on 24 molluscs from the northwestern Canadian Arctic Archipelago, which includes the northern and western coasts of Ellesmere Island. Reilly et al. (2019) argued that a ΔR of 770 years (using Marine13; Reimer et al., 2013) provided the best fit between a stacked paleosecular variation record



from Petermann Fjord sediments and a North Atlantic reference curve. This large ΔR was also consistent with large offsets
between radiocarbon dates on planktic and benthic foraminifera and ²¹⁰Pb derived chronologies for the past 100 years. In
summary, estimates used in the literature range from 0–770 years. In this study we have applied the new Marine20
calibration curve (Heaton et al., 2020), which results in ages that are ~150 years younger than equivalent Holocene ¹⁴C ages
calibrated using Marine09 or Marine13. Taking this into consideration we have applied a ΔR of 300±300 years. This broad
range provides a large uncertainty envelope. The upper bound approaches the older offset found by Reilly et al. (2019) for
Petermann Fjord (770 years using Marine13, which equates to ~620 years using Marine20) and the lower bound the 150
years (which equates to ~0 years using Marine20) commonly used to reconstruct terrestrial ice margins in the area (Kelly and
Bennike 1992; Young and Briner, 2015). The mean calibrated ages obtained using a ΔR of 300±300 years (equivalent to
~450 years using Marine13) provide a suitable estimate for sites that are influenced by Atlantic waters. This dependency on
water mass is one of the underlying problems in determining an applicable local reservoir correction. For example, the paired
benthic/planktic foraminifera samples we ran revealed offsets of 470-570 years in 6-GC and 7-PC which are currently bathed
in Atlantic waters, and 0 to 200 years at 10-GC which lies closer to the mixed surface layer (Fig. 3; Table 3). Additional
work is needed to resolve the issue of local marine reservoir offsets in the region, their dependency on water mass, and how
they have may have changed through time.

Table 3. Raw and calibrated radiocarbon dates and the type of material analysed. All data were calibrated using a ΔR 300 ± 300 years. Asterix highlight ages deemed outliers. All the outliers came from samples with low microfossil abundance where mixed benthic assemblages were dated.

| # | Lab ID | Sample ID | Depth (cm) | Comp. depth (cm) | ¹⁴C Age (a) | δ¹³C (‰) | Mean (cal a BP) | 1-σ (cal a BP) | Material dated |
|---|---|---|---|---|---|---|---|---|---|
| 1 | 152168 | 6-GC-2, 3-6 | 44.5 | 44.5 | 1730±75 | -0.3 | 850 | 540-1150 | *Cassidulina teretis* |
| 2 | 156282 | 6-GC-2, 70-75 | 112.5 | 112.5 | 4690±20 | 0.98 | 4330 | 3970-4770 | mixed benthic |
| 3 | 156283 | 6-GC-2, 93-95, A | 134 | 134 | 6870±30 | -0.13 | 6830 | 6510-7190 | mixed benthic |
| 4 | 156284 | 6-GC-2, 93-95, B | 134 | 134 | 6400±25 | 0.2 | 6330 | 5990-6680 | *Neogloboquadrina pachyderma* |
| 5 | 152196 | 6-GC-3, 101-105 | 295 | 295 | 8480±40 | -0.83 | 8550 | 8180-8920 | *Cassidulina teretis* |
| 6 | 152197 | 6-GC-CC | 494 | 494 | 9030±35 | -0.43 | 9180 | 8760-9550 | *Cassidulina teretis* |
| 7* | 156272 | 7-PC-1, 65-72 | 68.5 | 68.5 | 6790±35 | -0.67 | 6750 | 6440-7120 | mixed benthics |
| 8 | 152208 | 7-PC-1, 103-109 | 106 | 106 | 2670±20 | 0.33 | 1850 | 1470-2210 | mixed benthics |
| 9 | 156294 | 7-PC-1, 103-115 | 109 | 109 | 2620±45 | -0.12 | 1790 | 1410-2130 | mixed benthics |
| 10 | 152169 | 7-GC-2, 32-34 | 101 | 140 | 2890±50 | -0.62 | 2100 | 1720-2480 | *Cassidulina teretis* |



| 11 | 152198 | 7-TWC-CC | 97 | 164 | 3260±25 | -0.73 | 2540 | 2160-2930 | *Cassidulina teretis* |
| 12 | 156309 | 7-PC-2, 23-25 | 175 | 175 | 4060±30 | -0.54 | 3540 | 3150-3940 | mixed benthics |
| 13 | 152207 | 7-GC-2, 85-87 | 154 | 197 | 5940±25 | 1.2 | 5820 | 5520-6190 | *Cibicides lobatulus* |
| 14 | 152171 | 7-PC-2, 63-65, A | 215 | 215 | 7090±70 | -0.7 | 7050 | 6750-7420 | *Cassidulina teretis* |
| 15 | 153807 | 7-PC-2 63-65, B | 215 | 215 | 6520±35 | -0.54 | 6470 | 6140-6830 | *Neogloboquadrina pachyderma* |
| 16 | 156293 | 7-GC-3, 114-118 | 335 | 415 | 8060±85 | -1.62 | 8070 | 7710-8380 | mixed benthics |
| 17* | 156310 | 7-PC-3, 122-126 | 424 | 424 | 9270±45 | -1 | 9500 | 9100-9920 | mixed benthics |
| 18* | 156308 | 7-GC-3, 125-127 | 345 | 428 | 9350±45 | -1.16 | 9610 | 9250-10050 | mixed benthics |
| 19 | 152172 | 7-PC-4, 143-145 | 595.5 | 595.5 | 8410±100 | -0.91 | 8470 | 8070-8850 | *Cassidulina teretis* |
| 20 | 152170 | 7-GC-CC | 520 | 611 | 8900±100 | -1.06 | 9020 | 8630-9430 | *Cassidulina teretis* |
| 21 | 152209 | 7-PC-5, 103-105 | 706.5 | 706.5 | 9460±45 | -0.72 | 9750 | 9390-10170 | *Cassidulina teretis* |
| 22 | 152173 | 7-PC-5, 143-145 | 746.5 | 746.5 | 9210±95 | -0.72 | 9420 | 9010-9850 | *Cassidulina teretis* |
| 23* | 156295 | 7-PC-6, 76-78, A | 830.5 | 830.5 | 11000±160 | -0.96 | 11850 | 11410-12380 | mixed benthics |
| 24 | 152210 | 7-PC-6, 76-78, B | 830.5 | 830.5 | 10200±40 | -0.28 | 10730 | 10330-11160 | *Cassidulina teretis* |
| | | | | | | | | | |
| 25 | 154575 | 8-GC-2, 102-104 | 149 | 180 | 3290±25 | 1.46 | 2580 | 2200-2970 | *Cassidulina teretis* |
| 26 | 152211 | 8-GC-3, 102-104 | 301 | 341 | 4090±20 | | 3580 | 3190-3980 | *Cassidulina teretis* |
| 27 | 156285 | 8-PC-3, 61-63, B | 362.5 | 400.5 | 4480±20 | -0.04 | 4070 | 3680-4480 | *Cassidulina teretis* |
| 28* | 156311 | 8-PC-3, 61-63, A | 362.5 | 400.5 | 9280±35 | -0.31 | 9520 | 9120-9930 | mixed benthics |
| 29* | 156312 | 8-PC-3, 63-65 | 364.5 | 402.5 | 6310±30 | 0.32 | 6230 | 5900-6580 | mixed benthics |
| 30 | 156288 | 8-PC-3, 73-75 | 374.5 | 412.5 | 5000±20 | -1.03 | 4720 | 4370-5170 | mixed benthics |
| 31 | 152213 | 8-PC-3, 83-85 | 384.5 | 422.5 | 5260±30 | -0.27 | 5040 | 4710-5460 | *Cassidulina teretis* |
| 32 | 152212 | 8-GC-4, 62-64 | 411.5 | 495 | 7220±40 | -0.37 | 7190 | 6900-7540 | *Cassidulina teretis* |
| 33 | 152174 | 8-GC-CC | 499 | 609 | 7690±65 | -0.85 | 7680 | 7380-8000 | *Cassidulina teretis* |
| 34 | 152298 | 8-PC-5, 46-49 | 648 | 686 | 9140±40 | -0.75 | 9330 | 8940-9750 | *Cassidulina teretis* |
| 35* | 156313 | 8-PC-5, 68-72 | 670.5 | 708.5 | 10400±70 | -0.62 | 11010 | 10550-11450 | mixed benthics |
| | | | | | | | | | |
| 36 | 152279 | 9-TWC-CC | 101 | 201 | 3550±55 | -0.31 | 2900 | 2550-3320 | mixed benthics |
| 37 | 152300 | 9-PC-4, 103-105 | 556.5 | 556.5 | 4380±25 | -0.33 | 3940 | 3560-4350 | *Cassidulina teretis* |
| 38 | 152280 | 9-PC-4, 123-125 | 576.5 | 576.5 | 6620±65 | -0.47 | 6570 | 6250-6940 | *Cassidulina teretis* |
| 39 | 152299 | 9-GC-CC | 590 | 650 | 7780±40 | -0.5 | 7770 | 7460-8080 | *Cassidulina teretis* |
| 40 | 152281 | 9-PC-6, 53-55 | 810 | 810 | 8370±110 | -0.79 | 8420 | 8010-8780 | *Cassidulina teretis* |





| 41 | 156287 | 10-GC-1, 50-54 | 52 | 52 | 2450±20 | -0.05 | 1600 | 1250-1950 | mixed benthic |
|---|---|---|---|---|---|---|---|---|---|
| 42 | 152301 | 10-GC-1, 102-104, A | 103 | 103 | 6750±35 | -0.6 | 6710 | 6380-7070 | *Cassidulina teretis* |
| 43 | 153806 | 10-GC-1, 102-104, B | 103 | 103 | 6750±30 | 0.15 | 6710 | 6390-7070 | *Neogloboquadrina pachyderma* |
| 44 | 152302 | 10-GC-2, 132-134 | 279.5 | 279.5 | 7880±45 | -0.34 | 7880 | 7560-8190 | *Cassidulina teretis* |
| 45 | 152282 | 10-GC-CC, A | 288 | 288 | 8270±90 | -0.59 | 8310 | 7910-8640 | *Cassidulina teretis* |
| 46 | 153808 | 10-GC-CC, B | 288 | 288 | 8060±35 | -1.61 | 8060 | 7720-8370 | *Neogloboquadrina pachyderma* |
| 47 | 152303 | 11-GC-1, 12-14 | 13 | 13 | 7540±30 | -0.3 | 7520 | 7240-7860 | *Cassidulina teretis* |
| 48 | 152283 | 11-GC-1, 102-104 | 103 | 103 | 11150±250 | -0.24 | 12010 | 11580-12610 | *Cassidulina teretis* |

## 4. Results

### 4.1 Lithostratigraphic Units

Data from the MSCL, XRF-scanning data, and CT imaging are used to identify six major lithologic units that are correlated from the fjord mouth (10-GC) to the inner bathymetric sill lying seaward of the modern ice tongue (coress 8-PC and 9-PC) (Fig. 5). A sixth core, 11-GC, located on the outer sill in the shallowest water depth (208 m), did not contain the same lithostratigraphic units, and only the basal radiocarbon date is used as a constraint for Early Holocene glacial ice retreat.

*LU6*: The lowermost lithologic unit (LU6) was recovered in 7-PC, 8-PC and 9-PC. It has moderately high but variable Ca contents and is a coarse-grained, poorly sorted diamict containing abundant gravel (2-64 mm) through cobble-sized (64-256 mm) clasts with a high bulk density (> 2 g/cm$^3$) (Figs. 5, 6). The mass-specific magnetic susceptibility is considerably higher in LU6 compared to the overlying LU5, generally exceeding 80-100 x10$^{-8}$ m$^3$/kg. Less than 5 cm of this unit were recovered in 9-PC, with greater recovery in 7-PC (54 cm) and 8-PC (>135 cm). The upper boundary with LU5 is abrupt in 8-PC,

gradual in 7-PC. In 9-PC, the thin LU6 and lowermost LU5 sediments are laminated, deformed and contain two large dropstones, making it difficult to determine the nature of the boundary. The LU6/LU5 boundary is marked by a pronounced up-core decrease in bulk density and mass-specific magnetic susceptibility. Two subunits (LU6a and LU6b) are recognized in the longer sequences from 7-PC and 8-PC. These subunits are separated by a sharp boundary (Fig. 6). Sediments from LU6a have a lower bulk density (2.0-2.2 g/cm$^3$) compared to LU6b (2.2-2.5 g/cm$^3$) and lower clast abundance. LU6b is a

massive clast-supported diamict, while LU6a contains lenses or intervals of visibly deformed fine-grained layers suggesting deposition beneath grounded ice (Figs. 5, 6).



**Figure 5. Summary of stratigraphy and correlation of lithologic units between coring stations in Sherard Osborn Fjord. Locations and uncalibrated ages of radiocarbon dates are shown. Key horizons in LU5 and LU4 that can be correlated between 7-PC, 8-PC and 9-PC are illustrated by thin black lines. Detailed CT-images with corresponding lithologic unit boundaries are provided for each coring station in the supplementary information.**

*LU5:* The defining characteristic of LU5 are pronounced mm- to cm-scale laminations and absence of bioturbation. Laminae are sometimes visible in the split core sections as alternating reddish-brown and lighter tan-colored laminae (Fig. 6). The laminations are best defined by variations in Ca and bulk density. Ca enrichment arises from increased inputs of detrital carbonate, eroded from the surrounding Cambrian and Silurian bedrock. Where visible to the naked eye, the lighter tan-colored laminations are enriched in Ca and denser (lighter in CT-scanning images) (Fig. 5). Ca-enriched layers are skewed

towards medium and coarse silts, while laminae with lower Ca abundance are skewed towards fine silt and clay (Fig. 7). The higher bulk density of the light-colored laminae likely is caused by the coarser grain size and higher detrital carbonate



content, which has a higher grain density compared to quartz and clay minerals. LU5 has a low abundance of irregularly spaced clasts >1 mm that are more prevalent near the base of the unit (Fig. 5). In cores 10-GC, 7-PC, and 6-GC, there is a notable up-core decrease in the bulk density, Ca-content and laminae thickness of LU5. Mass-specific magnetic

susceptibility also increases up-core, as the relative Ca-abundance decreases, consistent with a larger contribution from siliciclastic sediments. In 6-GC and 7-PC, these long-term trends are interrupted by a 0.50-1.0 m thick interval with elevated Ca contents and thicker laminae, found near the top of the unit.

CT images illustrate that laminations in LU5 are wavy to lenticular in nature, and hence deposited in part by traction
processes under the influence of bottom water currents (Fig. 6). Throughout LU5, the laminations are disrupted by numerous small-scale normal faults. In 6-GC and 7-PC, faulting in LU5 is more pronounced than in the overlying laminated sequence of LU4 (Fig. 6). Laminations become less convoluted and more planer towards the top of LU5, indicating an overall reduction in traction dominated sedimentation through time.

At the two stations from the inner sill, 8-PC (238 mbsl) and 9-PC (274 mbsl), LU5 is considerably thinner than in 6-GC (633 mbsl) and 7-PC (559 mbsl), which lie further seaward, and in deeper water depths (Fig. 5). The thinner sequences of LU5 in 8-PC and 9-PC also contain intervals where the laminations appear to be truncated or eroded, sometimes with a lag deposit of coarse-grained material (*see* 9-PC-6 in Fig. 6), indicating that deposition of LU5 was discontinuous at these shallower water stations. On the outer sill, LU5 is also comparatively thin, containing highly fractured and wispy laminations that are
truncated or eroded in some intervals. This also indicates a more dynamic depositional environment, likely influenced by current activity or ice scouring. The base of LU5 in 10-GC is dominantly composed of sand and gravel sized material.

*LU4:* Faint, planar mm-scale laminations that lack notable evidence for bioturbation define LU4. The transition between LU5 and LU4 is gradual, with the base of LU4 being identified by a notable and correlative decline in Ca-abundance that is
identified at all the coring stations (Fig. 5). Laminations in LU4 are only weakly visible by the naked eye but are evident in the CT scanning data (Fig. 6). In 7-PC and 6-GC, laminations appear to transition from wavy to lenticular with frequent faulting in LU5, to planar in LU4, indicating an increased influence of suspension settling on sedimentation. With the exception of 10-GC on the outer sill, coarse ice-rafted clasts remain dispersed in low numbers through LU4. LU4 is a silty clay, containing very minor amounts of sand (<1%), with some samples showing a slight elevation of medium to coarse silt
(Fig. 7). The bulk density remains relatively constant (1.6-1.8 g/cm$^3$) with mass specific magnetic susceptibility higher than in LU5, and increasing up-core towards the top of the unit, consistent with a continued up-core decrease in detrital carbonate concentrations. Occasional minor pulses of Ca are seen throughout the unit. The LU5 to LU4 transition appears correlative across the fjord, and many small-scale features in the Ca, bulk density and magnetic susceptibility logs can be traced between the sites (Fig. 5). This suggests a uniform depositional environment across the fjord during this time interval.






*LU3:* Across Sherard Osborn Fjord, LU4 is capped by LU3, a second diamict containing large gravel to cobble-sized clasts (Figs. 5, 6). LU3 is recognized by its high bulk density (1.8-2.0 g/cm$^3$), elevated Ca-content and abundant clast content (10 - >100 clasts/cm$^3$) (Fig. 5). LU3 has a low mass specific magnetic susceptibility, except at 8-PC where cobble and bolder sized clasts of felsic rocks were recovered, which presumably led to the high susceptibility measurements. CT images from

6-GC and 7-PC show that the lower boundary is heavily bioturbated, while at 8-PC and 9-PC it is marked by a higher concentration of coarse clasts that tend to fine upwards (Fig. 6). At 8-PC the coarse clasts appear to have been smeared down the inside of the core liner, artificially increasing the clast content of the underlying LU4. In all cores, the lower boundary of LU3 and uppermost 5-10 cm of LU4 are bioturbated. The upper boundary of LU3 is relatively sharp at all sites. The matrix material of LU4 is clayey-silt, with a generally coarser mean grain size compared to LU4, and a larger sand-sized

contribution (up to 4.5%). The only indication of deformation is seen in 8-PC, surrounding the region where large dropstones were recovered. The absence of deformation at the other sites suggests that this is an artifact of coring and not a primary depositional feature. The absence of deformation indicates that this unit is a glaciomarine diamict, primarily composed of ice rafted material deposited by icebergs and sea ice.

*LU2:* A sharp but bioturbated boundary separates LU3 from LU2. The defining characteristic of LU2 is the presence of bioturbation and lack of preserved laminations. This unit exhibits very little variation in the bulk density or Ca content (Fig. 5), is composed primarily of fine silt and clay, and contains very few scattered ice-rafted clasts (Fig. 6).  The thickness of LU2 varies considerably across the fjord, with the thickest occurrences in 6-GC and 7-PC. LU2 is nearly absent in sediments from the outer sill (10-GC) (Fig. 5).


*LU1:* A return to a laminated facies lacking coarse ice-rafted clasts and bioturbation defines LU1. The grain size spectra of LU1 closely resemble those from LU4 (Fig. 7). The laminations in LU1 are also highly fractured and distorted along the sides of the core. This is attributed to coring disturbance and the soft to soupy nature of these near surface sediments. LU1 is divided into two subunits. LU1b is only found at stations 8 and 9 on the inner sill. It is a faintly laminated unit, with some

discrete intervals where the laminations are disturbed, sometimes by bioturbation. LU1b has low abundances of Ca, similar to what is found in LU2 and LU4 (Fig. 5). LU1a is marked by the emergence of more pronounced mm- to cm-scale laminations that are only seen clearly in the CT images (Fig. 6). LU1a is marked by increased Ca concentrations, which are most apparent in 9-PC and 7-PC (Fig. 4). At stations 10, 6 and 7, LU1a sits abruptly on top of LU2 (Fig. 5), while at stations 8 and 9, there is a gradual transition between LU1b and LU1a.






**Figure 6. Examples of CT-images and color photographs representative of the six lithologic units. The vertical scale for each image is a constant 50 cm. The structures seen on the CT-images and the color photographs are not identical because the imaging planes are slightly different. The CT-images are representative 2D slices of the core interior, whereas the photos show the surface of split**
**cores. White in CT-images corresponds to higher density zones, black to lower density zones. Composite depths for each interval are given in red.**

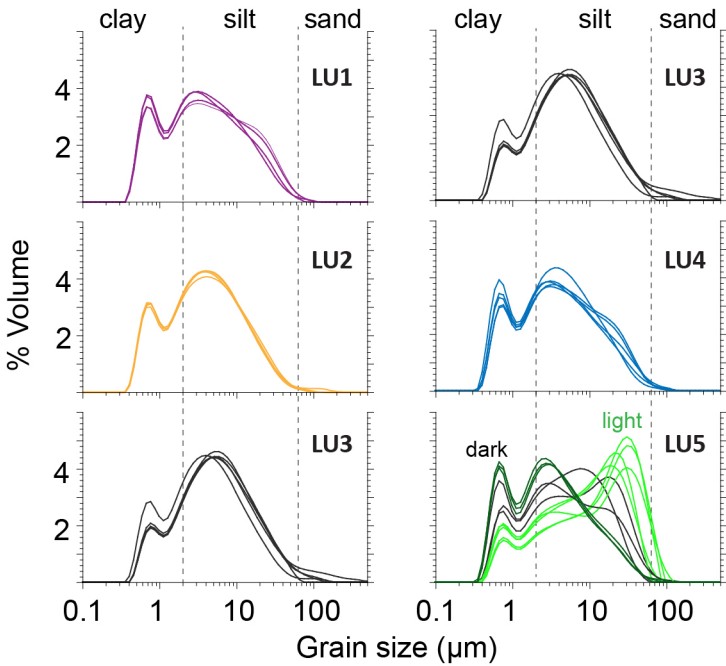

**Figure 7. Grain size spectra of lithologic units 1 – 5 illustrating the similar grain size distributions of sediments in LU1 and LU4, as well as the difference between the dark (less dense) and light (more dense) laminations in LU5.**

**4.1 Ages of unit boundaries**

Radiocarbon dating is used to constrain the ages of lithostratigraphic unit and subunit boundaries. In most cases samples were collected from within a few centimeters of these boundaries in one or more cores (Fig. 5). The youngest age underlying a unit or subunit boundary is used to date the boundary (Fig. 8). However, this is not possible for the LU5/LU6 and LU1b/LU2 boundaries. For LU6 the oldest age from the overlying unit is adopted as the boundary age. A younger than and older than age is provided for LU1b, based upon dates obtained from two different cores (Table 4). Of the 48 radiocarbon

dates, 7 were identified as outliers because the overlying ages are older and fell outside the 1σ calibrated age range. The lowermost sample in 8-PC (Sample #35, Table 3) is also considered an outlier, due to the substantially younger age returned in a sample 22.5 cm up-core (Sample #34, Table 3). LU5 sediments from this site were not deposited continuously, implying periods of erosion or non-deposition that could account for the large difference in ages between the lowermost dates in 8-PC.





The lithologic sequence from Sherard Osborn Fjord sediments spans nearly the entire Holocene (Table 4), with the oldest boundary (LU5/LU6) dated in 7-PC (Sample *#24*) to 10200±40 [14]C years (10330-11160 cal. a BP) while the youngest boundary (LU1a/2) is dated in 6-GC (Sample *#1*) to 1730±40 [14]C years (540-1150 cal. a BP). The oldest age comes from 11-GC (Sample *#48*, 11150±250 [14]C years) and provides a constraint on the timing for ice retreat from the outer sill.

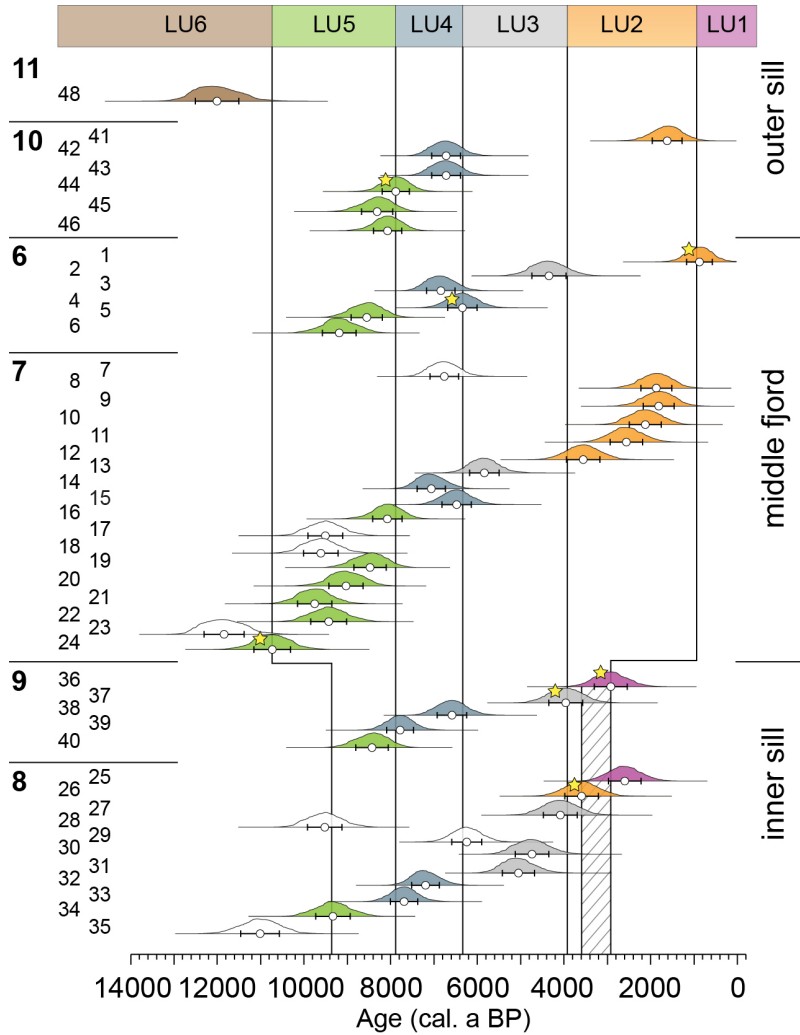

**Figure 8. Calibrated radiocarbon dates illustrating the mean and 1-σ age ranges. Color-coding is used to identify the lithologic unit that each date was acquired from. Dates from each core are presented in their stratigraphic order. White indicates ages that are deemed to be outliers. Yellow stars indicate dates used to define the ages of unit boundaries (Table 4). On the inner sill, the age range for the LU2–LU1b transition is bracketed by an 'older than' and 'less than' age from 9-PC and 8-PC respectively. Index numbers correspond to those in Tables 3 and 4. Date #47 from 11-GC is not shown as it could not be definitively placed within a**
**lithologic unit.**





**Table 4. Key radiocarbon dates and calibrated ranges defining basal ages of lithologic (sub)unit boundaries. Calibrated ages and uncertainty are rounded to 100 years, with the mean age from the calibrated age range reported. The first column refers to the radiocarbon date index number from Table 3.**

| # | Sample | Unit | $^{14}$C Age (a) | Age (cal ka BP) |
|---|--------|------|-----------------|-----------------|
| 1 | 06-GC-2, 3-6 cm | 1a | 1730±75 | 0.9 ± 0.3 |
| 36 | 9-TWC-CC | 1b | >3550±55 | 2.9 ± 0.4 |
| 26 | 8-GC-3, 102-104 | 1b | <4090±20 | 3.6 ± 0.4 |
| 37 | 9-PC-4, 103-105 cm | 2 | 4380±25 | 3.9 ± 0.4 |
| 4 | 06-GC-2, 93-95 cm, B | 3 | 6400±25 | 6.3 ± 0.3 |
| 44 | 10-GC-2, 132-134 cm | 4 | 7880±45 | 7.9 ± 0.3 |
| 24 | 7-PC-6, 76-78 cm, B | 5 (mid fjord) | 10200±40 | 10.7 ± 0.4 |
| 34 | 8-PC-5, 46-49 cm | 5 (inner sill) | 9140±40 | 9.3 ± 0.4 |

## 5. Discussion

### 5.1 Overview of the Holocene lithostratigraphic succession

The succession of stratigraphic units recovered in Sherard Osborn Fjord mirrors the classic deglacial facies transitions described from ice-shelf settings of Antarctica (Smith et al., 2019) and high-latitude northern hemisphere fjords occupied by marine terminating outlet glaciers and floating ice tongues (O'Cofaigh et al., 2001; Reilly et al., 2019) (Fig. 9). The interpretations of the stratigraphic units are here used to reconstruct the Ryder Glacier's dynamical history illustrated in Figure 10. The Sherard Osborn Fjord sequence involves a basal subglacial to glaciomarine diamict (LU6) overlain by a laminated meltwater dominated facies. The laminated facies fines upwards from a grounding zone proximal deposit with evidence for traction current activity (LU5) to a grounding zone distal facies deposited by suspension settling (LU4) beneath a floating ice tongue (Fig. 9).

The laminated grounding zone distal facies (LU4) is abruptly terminated by a clast-rich diamict (LU3). Similar facies transitions have been described as a response to migration of the ice shelf calving front, glacier surging or advance, or ice shelf collapse (Smith et al., 2019). In Sherard Osborn Fjord, the overlying sediments (LU2) are clast poor with extensive bioturbation suggesting deposition under more productive surface waters indicative of less extensive surface ice cover. As such, the facies succession, with LU3 separating the grounding zone distal (LU4) and more open-water bioturbated sediments (LU2), is consistent with a collapse of Ryder's ice tongue. The removal of an ice tongue marking the onset of LU3 allowed enhanced input of poorly sorted ice rafted material as debris-laden icebergs calved directly from the grounding zone were able to traverse the fjord. A return to a laminated facies (LU1) similar in character to LU4 indicates the successive re-



establishment of an ice tongue that eventually extended to the outer sill in Sherard Osborn Fjord. This basic interpretation is
supported by historical observations of Lauge Koch, who mapped the ice- tongue limit in proximity to the outer sill in 1917
(Koch, 1928) (Fig. 2).

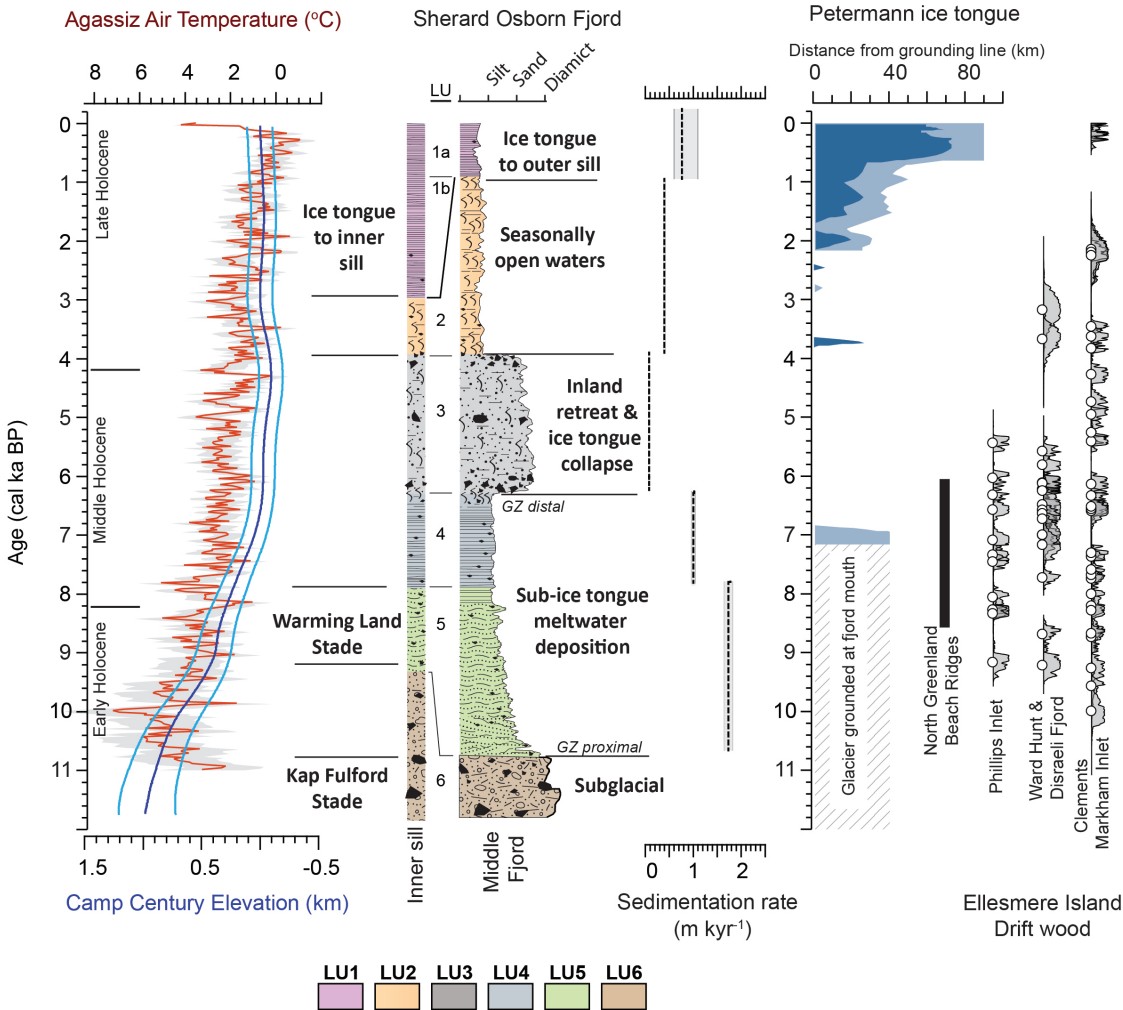

**Figure 9.** Representative lithostratigraphic column and environmental interpretations for the sedimentary cover of the inner sill
and middle fjord. Approximate linear sedimentation rates are provided for the middle fjord (station 7, Fig. 2) using the derived
unit boundary ages and their uncertainty (Table 3). The interpreted lithostratigraphic column is shown alongside the Agassiz air
temperature reconstruction and elevation changes of the GrIS at Camp Century (Lecavalier et al., 2017); the reconstructed extent
of Petermann Glacier ice tongue (Reilly et al., 2019); the period of beach ridge formation along the north coast of Greenland
indicating the absence of perennial land fast ice (Moller et al., 2010); and driftwood delivery to northern Ellesmere Island
(England et al., 2008).



## 5.2 Early Holocene glacier and ice tongue dynamics

The oldest recovered sediments come from 11-GC on the outer sill where a sample of the benthic foraminifera *Cassidulina teretis* returned a date of 12±0.5 ka. Due to the limited amount of sediment that has accumulated above the bedrock/till at 6-GC, 10-GC and 11-GC (Fig. 4), we interpret this as a minimum age for the retreat of Ryder Glacier from the outer sill, where
it was grounded 80-90 km seaward from its current position (Fig. 10). Deposition of the grounding zone proximal meltwater sediments (LU5) above subglacial diamicts in 7-PC commenced at 10.7±0.4 cal ka BP, indicating that the glacier had retreated further inland by this time (Fig. 10). The calibrated ranges of radiocarbon dates from the base of 7-PC (10.3-11.1 cal ka BP) conform to the reported age for the Kap Fulford Stade (>10.5 cal ka BP) (Kelly and Bennike, 1992) (Fig. 10). It is less certain if the older age from 11-GC on the outer sill (11.6-12.6 cal ka BP) (Fig. 8, Table 3) implies that grounded ice
was here during the Kap Fullford Stade or an earlier stage of deglaciation.

Following retreat of Kap Fulford glacial ice, the sedimentary sequence is dominated by the Ca-rich laminated meltwater facies (LU5). The coarser grained Ca-enriched laminae are associated with increased meltwater input from glacial erosion of the surrounding Cambrian and Silurian carbonate shelf bedrock (Fig. 2). The strong meltwater signal captured in the LU5
coincides with the rapid Early Holocene reduction in the height of the Greenland ice sheet (Lecavalier et al., 2017) (Fig. 9).

Sediments from LU5 are comparatively thin at the inner sill coring sites, and difficult to directly correlate with the records from 6-GC and 7-PC. Erosional events identified in LU5 sediments from the inner sill suggest discontinuous sedimentation (*see* 9PC-6 Fig. 6). Sitting in much shallower water depths (228-271 mbsl; Table 1), glacial ice likely remained grounded
here after it retreated from stations 6 and 7. Deposition on the inner sill would also have been influenced by bottom scouring currents and debris flows while the glacier was grounded (or near flotation) on shallow regions of the sill that rise to depths of 193 m. Therefore, we suggest that Ryder Glacier remained grounded on the inner sill until sometime after 9.3±0.4 cal ka BP, which marks the onset of continuous sedimentation of LU5 at this location (Fig. 10; Table 4). This is consistent with the age of the Warming Land Stade, bracketed between >9.5 cal ka BP and 8.0 cal ka BP by Kelly and Bennike (1992). The
shallow inner sill, first identified through bathymetric mapping during the *Ryder 2019* expedition (Jakobsson et al., 2020), is very close to the previously inferred grounding zone location for the Warming Land Stade and provides a natural pinning point for the marine-based glacier during this standstill (Fig. 2).

In 6-GC and 7-PC there is a broad period of elevated detrital carbonate delivery (high Ca-content) that occurs near the top of
LU5 (Fig 5). This likely reflects enhanced meltwater delivery during terrestrial ice retreat at the end of the Warming Land Stade prior to the LU4/LU5 boundary at 7.6±0.40 cal ka BP (Fig. 10; Table 4). Across Sherard Osborn Fjord, deposition of the laminated LU4 sediments continued until 6.3±0.3 cal ka BP (Fig. 10; Table 4), which is very similar to the age (6.0 cal ka BP) for the minimum Holocene extent reported for Ryder Glacier (Kelly and Bennike, 1992).





**Figure 10. Generalised lithology and corresponding reconstructions of Ryder Glacier and ice tongue during deposition of the lithologic units. The profile location is illustrated in Fig. 11. The bathymetry was re-gridded from BedMachine v3 (Morlighem et al., 2017) after adding bathymetric data collected on *Ryder 2019*. Substantial modifications to the former digital elevation model exist around the now mapped inner sill and beneath the modern ice tongue (see supplementary material for details). Yellow dots indicate approximate positions of coring stations discussed in the text.**





### 5.3 Middle Holocene inland retreat and collapse of Ryder's ice tongue

The second diamict facies (LU3) in the stratigraphic succession separates the faintly laminated sub-ice tongue sediments of LU4 and the bioturbated facies of LU2. As this unit is found from the inner sill all the way to the outer sill, and does not appear to be time-transgressive, it cannot be attributed to a slow landward migration of the ice tongue calving margin.

Instead, it is interpreted as a collapse facies (Smith et al., 2019) associated with the sudden disintegration of Ryder's ice tongue. While this interpretation fits with the overall facies succession, the ~2.4 kyr duration of LU3 (from 6.3±0.3 to 3.9 ± 0.4 cal ka BP; Table 4) is too long to be a simple abrupt collapse event, as these are generally associated with increased sediment accumulation rates (Smith et al., 2019).

An alternate interpretation is that LU3 sediments represent an erosional event. However, this does not fit with the overlapping ages of sediments from the top of LU4 at coring stations 6, 7, 9 and 10, and to a large degree the ascending ages found downcore through LU3 at station 8 (Fig. 8). Therefore, the widespread distribution, long duration, consistent boundary ages of LU3, and lack of evidence for pervasive ice scouring makes it hard to reconcile this facies with a glacial surge. This unit is also much younger than the inferred age for the drainage of the ice dammed lake that developed on Wulff Land

between Ryder and C. H. Ostenfeld glaciers between 7.9-8.5 cal ka BP (Kelly and Bennike, 1992). These observations suggest that LU3 marks a period of low sedimentation that began around 6.3±0.3 cal ka BP when terrestrial studies suggest that Ryder Glacier had retreated further inland than its current position (Kelly and Bennike, 1992).

The explanation that best fits evidence from terrestrial field studies, and the overall facies succession, is that the condensed

diamict of LU3 was deposited when Ryder Glacier retreated far enough inland to become cut-off from the main fjord. In Sherard Osborn Fjord, a relatively deep, isolated marine embayment exists behind a prominent topographic high lying 40 km inland of the modern grounding zone (Fig. 11). Here elevations increase to between 100-200 m above sea level, with elevations surrounding the embayment reaching 400 – 600 m above sea level (Figs. 10, 11). Retreat of the grounding zone onto this topographic high (possibly a former grounding zone wedge), or even inland of it, would explain the apparent rapid

disintegration of the ice shelf at the end of LU4. The entrainment of meltwater-derived sediments within the isolated embayment or even behind the inner sill, can account for the overall reduction in sedimentation rates in the outer fjord. This is consistent with observations from modern proglacial lakes that illustrate their efficiency at disrupting meltwater fluxes and sequestering sediments (Carrivick and Tweed, 2013; Bogen et al., 2015; Piret et al., 2021). While grounded in this distal position, calving could have discharged some icebergs into Sherard Osborn Fjord, delivering coarse-grained ice rafted

sediments. Elevated Ca contents within LU3 (Fig. 5) attest to sediment provenance being primarily from landward of the inner sill (Fig. 2). Furthermore, the sedimentation rates during LU3 are low (9-10 cm/kyr; Fig. 9) and comparable with sedimentation rates of 10-30 cm/kyr reported by Dowdeswell et al. (1994) for massive diamicts derived from iceberg rafting in Scoresby Sund and the adjacent East Greenland shelf. A final indication that Ryder Glacier retreated to the innermost



fjord or beyond is the lack of iceberg scouring seen in bathymetric data on the inner sill. This suggests that the calving front
was grounded in water shallow enough to not produce icebergs in excess of 190-200 m thickness – a condition that can
easily be met by the landward retreat of Ryder Glacier (Fig. 10).

A modern analogue for this configuration is seen in neighboring Victoria Fjord. Today C. H. Ostenfeld Glacier is grounded
on a topographic high at the end of the fjord, in an equivalent position to the landward terminus of Sherard Osborn Fjord
where the proposed Middle Holocene ice margin is located (Fig. 10). Between 2000 and 2006, the C. H. Ostenfeld ice
tongue, which protruded from a narrow over-deepened marine channel, largely disintegrated (Moon and Joughin, 2008),
while the grounding zone may soon become increasingly land-based and discharge grounded ice directly into Victoria Fjord
(Hill et al., 2017) (Fig. 10).

The continued retreat of Ryder Glacier through the Early and Middle Holocene is consistent with many regional
paleoclimate proxies from around northern Greenland. This includes beach ridge formation along the north Greenland coast,
indicating more open water conditions that persisted until ~6 cal ka BP (Möller et al., 2010; Funder et al., 2011), reduced
winter sea-ice conditions in Bliss Lake until 6.5 cal ka BP (Olsen et al., 2012), and peak late summer air temperatures
inferred from $\delta^{18}$O of chironomids in Secret and Deltasø lakes that were 2.5-4°C warmer then present until 6.2-6.0 cal ka BP
(Axford et al., 2019; Lasher et al., 2017) (Fig. 1). The paleotemperature estimates from these lakes indicate a slight cooling,
but persistent, stable and positive temperatures between ~6.0-4.0 cal ka BP (Lasher et al., 2017; McFarlin et al., 2018).
Diminished sediment delivery due to the presumably stable inland position of Ryder Glacier lasted from 6.3±0.3 to 3.9±0.4
cal ka BP (Table 4). The inferred re-advance into Sherard Osborn Fjord, marked by increased sediment delivery at the onset
of LU2, occurred around 3.9±0.4 cal ka BP. This timing for glacier advance is consistent with cooling seen in lake based
temperature reconstructions around 4 cal ka BP (Lasher et al., 2017) and the oldest estimated age (3.5 to 4.0 cal ka BP) for
ice at the base of the southern dome of Hans Tausen ice cap, which had disappeared during the Middle Holocene (Madsen
and Thorsteinsson, 2001; Landvik et al., 2001; Zekollari et al., 2017).

A stable inland position for Ryder Glacier during LU3 fits with generally colder conditions that prevailed over north
Greenland towards the end of the middle Holocene. This is reflected in the development of more prolonged winter ice cover
in Bliss Lake after 5.9 cal ka BP (Olsen et al., 2012) and a period of low melt rates for the north Greenland ice sheet inferred
from elevation changes at Camp Century (Lecavalier et al., 2017) (Fig. 9). Therefore the prolonged slow deposition of coarse
IRD during LU3 is likely a combined effect of limited but persistent iceberg calving, the slow melt-out of englacial material
from the disintegrated ice tongue, as well as continued input of sea-ice rafted material entrained during periods of shore-fast
sea ice growth and decay.



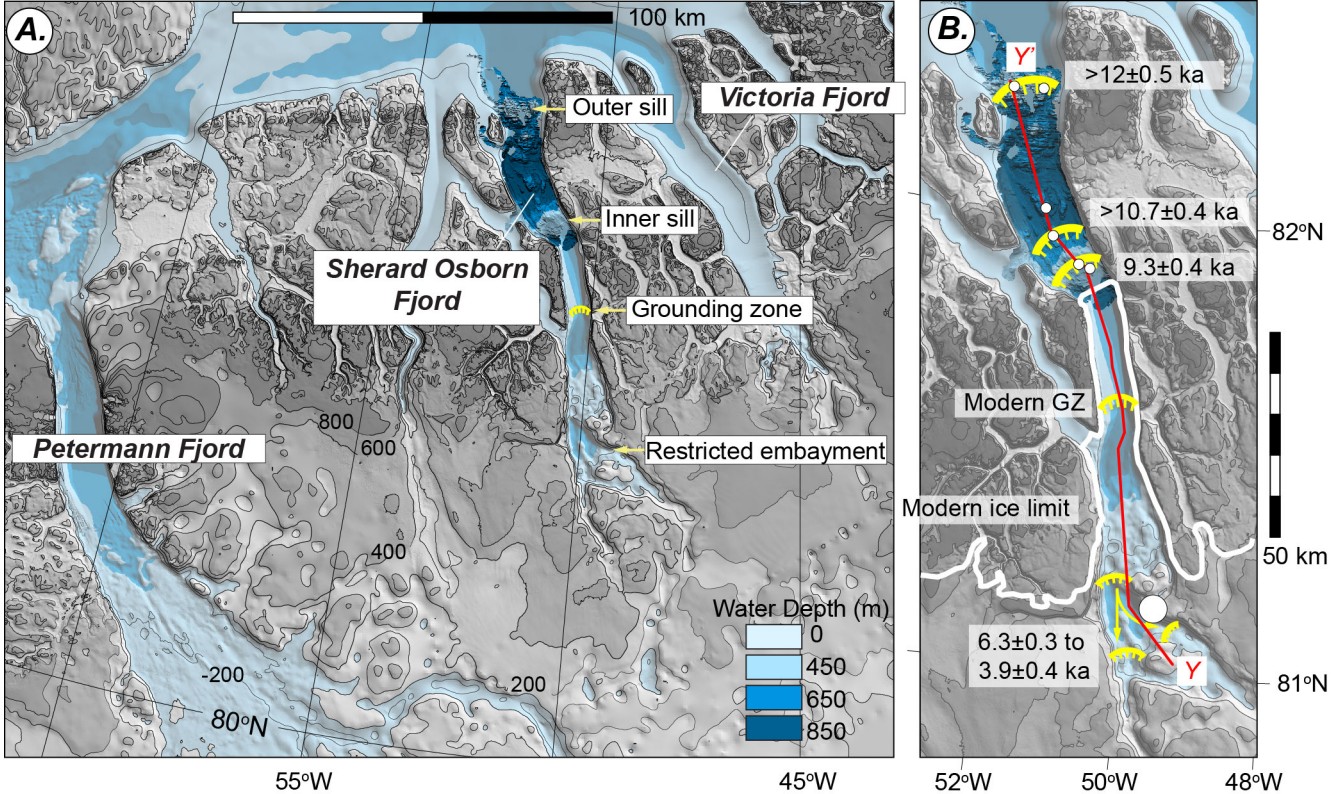

**Figure 11. A. Bathymetry and sub-glacial topography of north Greenland from Morlighem et al. (2017) illustrating pronounced differences between the broad submarine trough that extends inland from Petermann Glacier and the more restricted, and rapidly shoaling topography in Sherard Osborn and Victoria Fjords. B. Detail of Sherard Osborn Fjord showing the two mapped sills that correspond to the likely grounded limits of Ryder Glacier during the Kap Fulford and Warming Land Stade. At the end of the Middle Holocene, Ryder Glacier likely retreated to a more inland position becoming largely land-based and cut off from the main fjord by the restricted embayment. White dots are the coring sites discussed in the text. Red line (Y-Y') shows the position of the bathymetric profile used in Fig. 10 (additional detail in the supplementary information). GZ = Grounding Zone.**

**5.4 Late Holocene cooling and the re-growth of Ryder's ice tongue**

The advance of Ryder Glacier and the re-growth of local ice caps surrounding Sherard Osborn Fjord is described by the Steensby Stade re-advance. Its onset is not well constrained by terrestrial field data, but is believed to have started sometime after 5.1-4.7 ka (Kelly and Bennike, 1992), with outlet glaciers reaching their maximum extents by the start of the 1900's. Thus, in marine records from Sherard Osborn Fjord, the Steensby Stade encompasses deposition of LUs 2 and 1.

The highly bioturbated LU2 sediments appear abruptly on top of LU3, with the onset of deposition sometime after $3.9 \pm 0.4$ cal ka BP (Table 4). While the inland glacier may have started advancing earlier, increased sedimentation rates at the onset





of LU2 (Fig. 9) likely marks the seaward migration of Ryder Glacier to a marine-based position in the innermost fjord. Within this setting, the re-growth of a limited ice tongue would act as a filter for coarse iceberg rafted sediments and can explain the paucity of IRD in LU2 sediments. Extensive bioturbation in LU2 attests to warmer climatic conditions and the

absence of an ice tongue seaward of the mapped inner sill and likely less persistent sea-ice cover.

On the inner sill, deposition of laminated sediments lacking bioturbation (LU1b) marks the re-growth of the ice tongue out to this location (Fig. 10) between 3.6 ± 0.4 and 2.9 ± 0.4 cal ka BP. This is roughly 300-1000 years after the glacier became marine-based at the end of LU3. The persistence of bioturbation (LU2) at stations in the middle fjord (stations 6 and 7),

indicates an earlier establishment of the ice tongue over the inner sill, while more open water conditions persisted further seaward (Fig. 9). This configuration is similar to what we found during our expedition in 2019 (Fig. 2). It appears that this was a relatively stable configuration that persisted for another 2-2.7 kyrs (until 0.9 ± 0.3 cal ka BP; Table 4), before the ice shelf extended towards the outer sill and laminated LU1a sediments were deposited in the middle fjord (Figs. 9, 10). The long-term stability of the ice tongue terminus near the inner sill is consistent with the inner sill acting as a barrier for Atlantic

water invasion and limiting basal ice tongue melting (Jakobsson et al., 2020).

The Late Holocene re-growth of Ryder's ice tongue has some parallels with the Petermann ice tongue, which began to reform between 1.9-2.3 cal ka BP, after being absent since its collapse around 6.9 cal ka BP (Reilly et al., 2019). Petermann also attained a stable ice tongue with an extent similar to 20[th] century historical observations between 0.4-0.9 cal ka BP

(Reilly et al., 2019). At both Ryder and Petermann the growth of ice tongues towards the outer fjords occurred much later than the establishment of multi-year landfast sea ice in front of Phillips Inlet and Disraeli Fjord on northern Ellesmere Island around 5.5 cal ka BP (England et al., 2008). On the other hand, ice tongue re-growth in Petermann (1.9-2.3 cal ka BP) and growth of Ryder's ice tongue to the inner sill in Sherard Osborn Fjord (by 2.9 ± 0.4 cal ka BP) are consistent with the development of more extensive sea ice around the northern Greenland margin by 2.5 cal ka BP (Funder et al., 2001) and only

intermittent periods of sea-ice free conditions in the Lincoln Sea after 3.9 cal ka BP based on the cessation of driftwood delivery to Clements Markham Inlet (England et al., 2008) (Fig. 9).

Uncertainty in the local reservoir correction leaves the timing for the maximum Late Holocene growth of Ryder's ice tongue poorly constrained. However, as with the final growth of Petermann's ice tongue (Reilly et al., 2019), it appears consistent

with numerous regional paleoclimate records that document the development of cooler conditions close to the transition from the Medieval Warm Period to the Little Ice Age. For example, geochemical proxies from Bliss Lake (Fig. 1) indicate the onset of cooling around 850 cal. a BP (Olsen et al., 2012), which agrees with the onset of cooler air temperatures inferred from the δ[18]O record of the Hans Tausen ice cap around 780 cal. a BP (Olsen et al., 2012; Hammer et al., 2001).





The absence of a thin bioturbated facies reflecting deposition since the 20[th] century retreat of Ryder's ice tongue is likely due
to incomplete recovery of surface sediments in the studied cores. Given the predicted range of sedimentation rates for LU1a
in 7PC (51-102 cm/kyr, Fig. 9), only 4– 8 cm of sediment would have accumulated since the retreat of Ryder's ice tongue to
its current position ~80 years ago. This thin sedimentary cover could easily have been missed during coring, especially
considering the soft nature of surface sediments encountered in the fjord.

**5.5 Controls on the Holocene dynamics of Ryder Glacier**

The retreat of Ryder Glacier from the fjord mouth (>10.7±0.4 cal ka BP) is consistent with the demise of glacial ice across
much of northern Greenland in the Early Holocene (Bennike and Björck, 2002; Larsen et al., 2010; Strunk et al., 2018;
Larsen et al., 2018). During the early stages of deglaciation, the Lincoln Sea was covered by thick shelf ice, fed by
coalescent glaciers emanating from Greenland and Ellesmere Island (England et al., 1999; Larsen et al., 2010). This eastward
flowing shelf ice overrode parts of the North Greenland coast (Johannes V. Jensen Land; Fig. 1) with its final break up
occurring between 10.3 - 10.1 cal ka BP (Larsen et al., 2010). Reduced sea-ice conditions in the Lincoln Sea at this time are
evidenced by the onset and persistence of driftwood delivery to Clements Markham Inlet (Fig. 9; England et al., 2008) after
glacial ice had retreated from the northernmost Nares Strait >10.1 ka (England et al., 1999). Future work on sediment cores
from *Ryder 2019* utilizing paleoenvironmental proxies for sea ice and oceanographic conditions may help resolve how
atmospheric warming (Cook et al., 2019) and warm Atlantic water advection (Wood et al., 2021) into the Lincoln Sea region
influenced Holocene sea-ice dynamics and glacier retreat.

Ryder Glacier retreated much earlier than Petermann Glacier to the south (Jakobsson et al., 2018; Reilly et al., 2019).
Petermann Glacier remained grounded at the mouth of Petermann Fjord, bordering Hall Basin, until 7.5 cal ka BP
(Jakobsson et al., 2018) even after Ryder had retreated from the inner sill and was >40 km from the mouth of Sherard
Osborn Fjord. In this respect, it is important to acknowledge that Petermann's retreat from the outer fjord was one of the
final events that occurred during deglaciation of Nares Strait and specifically of Hall Basin (Jakobsson et al., 2018). Glacial
ice in Hall Basin had remained in an advanced position abutting Robeson Channel until 9.3 cal ka BP (Fig., 1; Jakobsson et
al. 2018; England et al 1999). The ice margin had receded to the mouth of Petermann Fjord by 8.7 cal ka BP, prior to the
opening of Nares Strait between 8 and 8.5 cal ka BP (Jennings et al., 2011, 2019; Georgiadis et al., 2018). Therefore,
differences in the timing of retreat between Petermann and Ryder Glaciers appears closely tied to local glacial and sea-ice
conditions. These appear to have been less severe and more mobile in the Lincoln Sea in the Early Holocene compared to
southern Nares Strait region.

The sedimentary facies from Sherard Osborn Fjord documents seemingly gradual retreat and regrowth phases for Ryder
Glacier as evidenced by the generally low abundance of coarse IRD (Fig. 9) and the absence of lenses, or pulses of IRD that
would otherwise indicate ice tongue collapse events. For example, in Petermann Fjord the collapse of the ice shelf at 6.9 cal





ka BP is marked in sedimentary records by the abrupt appearance of IRD clasts in sediments from across the fjord (Reilly et al., 2019). The stability of Ryder Glacier's ice tongue is most likely related to the physiography of Sherard Osborn Fjord.


The inner sill in Sherard Osborn Fjord not only acts as a natural pinning point for glacial ice during its retreat but also impedes the flow of warm Atlantic subsurface waters into the innermost fjord (Fig. 3; Jakobsson et al., 2020). This likely reduces the subaqueous melting at the grounding zone and of the ice tongue today, and may have done so efficiently for much of the Holocene. The subglacial bed topography and physiography of the inner fjord will also exert an influence on 615 glacier (and hence ice tongue) stability. There appears to be a slight retrograde slope between the inner sill and modern grounding zone (Fig. 10), which are common in northern Greenland fjords and can be conducive to rapid glacial retreat (Enderlin et al., 2013; Carr et al., 2015; Hill et al., 2018). However, Sherard Osborn Fjord also narrows considerably landward of the inner sill and remains flanked by steep-sided cliffs (Fig. 11). Jamieson et al. (2012) have shown that even on a retrograde slope, retreating ice streams slow or can become pinned where they pass narrow sections in surrounding 620 topography. The sedimentary record from seaward of the inner sill contains no evidence for massive calving events that would otherwise suggest a collapse of the ice tongue during a rapid retreat down a retrograde slope. However, there is tremendous uncertainty regarding the bathymetry beneath the current ice tongue as no direct measurements exist, so even the presence of a retrograde slope is not certain. A more certain feature of the bathymetry is the steeply rising bed topography landward of the modern grounding zone (Figs. 10, 11). This feature, which could be a former grounding zone, may have 625 promoted the gradual retreat of Ryder Glacier and the stability of its ice tongue during the Early and Middle Holocene (Powell, 1990; Alley, 1991; Hill et al., 2018). A critical observation is that despite the apparent persistence of Ryder's ice tongue, the physiographic setting of Sherard Osborn Fjord, which appears to be conducive to glacier stability, did not halt the overall retreat of Ryder Glacier under the relatively mild changes in climate forcing during the Holocene.

From the results of this study, it is not possible to constrain the amount of inland retreat Ryder Glacier experienced during the Middle Holocene. Based on the cessation of sedimentation in the outer fjord, we suggest that it may have retreated at least into the restricted embayment at the end of Sherard Osborn Fjord (Figs. 10, 11). Unlike Petermann Glacier, where the bed topography widens and extends far beneath the GrIS at elevations below modern sea level, the bed of Ryder Glacier abruptly rises above sea level some 60 km landward of the modern grounding line (Fig. 11). This is comparable to the 635 smaller glaciers in the immediate surroundings and to the Køge Bugt glacier complex in southeast Greenland, where the steep land-ocean boundary limited the Middle Holocene retreat of the glaciers in this area (Dyke *et al.*, 2017). This suggests that the land-ocean interface might have been an important boundary that mediated or halted the Holocene retreat of many glaciers that are not located in deep troughs that extend beneath the inner ice sheet. Overall, our constrains on the Holocene dynamics of Ryder Glacier suggest that it could retreat another 40-60 km inland if climatic conditions remain similar or 640 exceed those of the Middle Holocene.



## 6. Conclusions

Like much of northern Greenland, Ryder Glacier responded acutely to climate variability in the Holocene. During the Early and Middle Holocene it retreated over 120 kilometers from a grounded position near the mouth of Sherard Osborn Fjord (80 km seaward of the modern grounding zone) to likely become land-based more than 40-60 km landward of its current

position by 6.3±0.3 cal ka BP. Throughout this long period of retreat, deposition of laminated, clast poor sediments attest to strong meltwater inputs and an overall stable ice tongue. Ryder Glacier remained land-based until the Late Holocene (3.9 ± 0.4 cal ka BP). As it again advanced into Sherard Osborn Fjord, an ice tongue developed that quickly grew out to its 21st century position near a prominent bathymetric sill, located 30 km seaward of the modern grounding zone. Today this sill reduces the incursion of Atlantic waters into the inner fjord and shields the grounding zone and ice tongue from basal

melting (Jakobsson et al., 2020). This was also likely the case for 2.0-2.7 kyrs in the late Holocene, before the ice tongue grew to its historical maximum extent around 0.9±0.3 cal ka BP. Late Holocene bioturbated sediments that are nearly devoid of ice rafted debris were deposited between 3.9±0.4 and 0.9±0.3 cal ka BP and indicate reduced ice cover over much of the middle and outer fjord. This contrasts with the near permanent shorefast ice conditions that existed in Sherard Osborn Fjord for much of the late 20th century (Higgins, 1989), but it is similar to conditions during the summer of 2019. The

physiography of Sherard Osborn Fjord appears to have had a stabilizing effect on the ice tongue of Ryder Glacier through the Holocene. Nevertheless, Ryder Glacier still retreated >40-60 km landward of its current position and remained there under the relatively mild climate forcing in the Middle Holocene, suggesting that it may again retreat completely from the fjord to an inland position if climate warming continues at its current pace.

### Data availability

Upon publication, the data presented in this manuscript will be archived in the Bolin Centre Database at the Bert Bolin Centre for Climate Research (https://bolin.su.se/data/).

### Author contributions

This manuscript was written by MO with subsequent input from all co-authors. MJ and LM led the Ryder 2019 Expedition. Cores were collected and processed shipboard by MO, TC, BR, GW, and LG. Shore-based analyses were conducted by MO,

BR, FV, CP, GW, AKO, OLM, AG, MM and HD. CS and JN collected, processed and interpreted the oceanographic data. MM provided the improved DEM for Sherard Osborn Fjord.

### Competing interests

The authors declare that they have no conflict of interest.



**Acknowledgments**

We are grateful to Captain and crew of Icebreaker *Oden* and personnel from the Swedish Polar Research Secretariat for their
support during the planning and execution of the Ryder 2019 Expedition. This expedition was endorsed as Explorer's Club
Flag Expedition #51. We thank the Swedish Polar Research Secretariat, Center for Coastal and Ocean Mapping, University
of New Hampshire and Stockholm University for supporting the *Ryder 2019* Expedition financially. M.O., M.J, C.S and
colleagues from Stockholm University were supported by grants from the Swedish Research Council (VR; grants 2016-
05092, 2016-04021, 2018-04350). T.C. and L.G. were supported by the USGS Land Change Program. Any use of trade,
firm, or product names is for descriptive purposes only and does not imply endorsement by the U.S. Government. We thank
June Padman (Oregon State University) and Carina Johansson (Stockholm University) for their help in shipboard core
processing.

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
