# Peer review of "The Holocene dynamics of Ryder Glacier and ice tongue in north Greenland"

_The Cryosphere, 2021_

## Author Comment (AC1)

**Responses to reviewer #1**

Thank-you for the positive comments and pointing out the grammatical and linguistic changes that could improve the manuscript. Below we outline how we have/will address each of these.

1. Lines 85 to 90. Can you just say how long the fjord is and how long the ice tongue is? These details can be stated more clearly.

> This depends on where we place the landward, sub-ice limit of the fjord. This is not 100% straightforward. We had originally referenced the length of the fjord with respect to the ice tongue terminus: *"Sherard Osborn Fjord is ~17 km wide and extends ~55 km from the ice tongue margin of Ryder Glacier out towards the Lincoln Sea. Ryder Glacier is currently grounded below sea level, with the grounding zone located ~26 km landward of the ice tongue terminus"*
>
> However, we agree that this could be clearer, and can use the modern grounding zone position as a reference. Therefore we can write:
> *"Sherard Osborn Fjord is ~17 km wide and extends ~81 km from the modern grounding zone of Ryder Glacier out towards the Lincoln Sea. Ryder Glacier is currently grounded below sea level, with an ice tongue that extends ~55 km from the grounding line out into the fjord"*

2. Line 88. Instead of sills 'dissecting' perhaps say 'crossing'. I don't think dissecting is quite right.
> *We have made this change.*

3. Line 89. Define what you mean by 'overdeepened'.
> *'overdeepened' has been removed. The sentence now reads: "These sills bound a basin that has a maximum depth of 890 m"*

4. Line 141. Delete 'that extent'.
> *Yes, this has been deleted. Not sure why it was there in the first place.*

5. Line 150. Replace 'exerting' with another word…'exhibiting'?
> *Changed as suggested.*

6. Line 162. Delete 'a'.
> *Yes, this is deleted.*

7. Line 167. Replace 'highly lithified' with 'compacted' or 'consolidated'. It has not been formed into rock so is not lithified.
> *We have changed to 'consolidated'*

8. Line 214. Delete 'the'.
> *Yes, we deleted this.*

9. Line 221. Is the piston core just the 'reference core'? I don't know why it is called 'undistorted'. That seems unlikely actually, and the word is not needed.
> *The reviewer is correct, the word is not necessary and slightly misleading. We have removed 'undistorted'.*

10. Line 229 under radiocarbon dating. I suggest you *use Cassidulina neoteretis* throughout and cite Cage et al., 2021 https://doi.org/10.5194/jm-40-37-2021 which is a paper that clearly shows how to identify C. neoteretis and C. teretis.

> *We have changed the name throughout the paper to Cassidulina neoteretis and refer the readers to Cronin et al. (2019) and Cage et al. (2021) for discussions of this foraminiferal species in the Arctic and Nordic seas respectively.*

11. Line 230. What benthic foram species were included in the mixed benthics. These appear to be older than the single species dates on C. neoteretis. It is important to present the species dated. If Miliolid species were included in the dated material (e.g. Triloculina or Quinqueloculina) this can explain the too old results. Hopefully the specific contents do the mixed benthic dates was recorded and can be reported here. It is useful information to guide future chronological studies.

> *Although Miliolid species were present, they were not used in the mixed benthic dating. We have added what species were included in the mixed benthic samples. These include C. neoteretis, C. reniforme, O. tener, E. excavatum clavata. We have included this in the manuscript. Unfortunately, we do not have details on the exact composition in each sample.*

12. Line 260. Neoteretis.
> *This has been corrected throughout Table and C. neoteretis is now used.*

13. Line 266. Delete double s in cores.
> *Spelling mistake is fixed.*

14. Line 271. Is it diamict or diamicton. I think diamicton is correct.
> *We have changed to 'diamicton' throughout.*

15. Line 280. Is deformation beneath grounded ice the only way to get deformation? Can this deformation be due to coring or slumping? I am not contesting that the unit is subglacial in origin.

> *The reviewer is correct in that deformation in sediment cores can occur from numerous causes including coring deformation and mass transport/gravity flow deposits or glacial deformation. In this case, it does not appear to be coring induced, as the interval is found in the middle of a core section and has abrupt upper and lower contacts with laminated sediments and a massive clast-supported diamicton respectively. While a transition from a massive diamicton, to a deformation till to grounding-zone proximal laminated meltwater influenced sediments makes perfect sense, we cannot rule out gravity-driven deposition.*

> *In the revised manuscript we can remove the interpretation from this sentence and include it at the end of the paragraph (as reviewer 2 suggested). In doing so we will acknowledge that it can either have been deposited beneath grounded ice (deformation till), or proximal to the grounding line (gravity flow deposit). This will not influence our glacial reconstructions in any way, but is a more honest interpretation of the data.*

16. Line 326.  Suggest you delete 'Across Sherard Osborn Fjord' and just begin the sentence with LU4.

*This has been changed accordingly.*

17. Or you could say 'Throughout Sherard....'.

*We have changed 'Across' to 'throughout'.*

18. Line 430.  Delete one l in Fulford.

*Corrected spelling mistake.*

19. Line 480.  Not clear what 'become cut-off from the main fjord' means.  Does it mean the ice retreated onshore?

*We had originally followed this sentence with a more detailed explanation of what we meant. However, We can improve the clarity by simply removing 'far enough inland to become cut-off from the main fjord ' since the following 3-4 sentences describe how we believe the inland retreat would result in the slow deposition of the diamicton of LU3.*

*Therefore the new opening sentences of this paragraph will be changed from:*
 *"The explanation that best fits evidence from terrestrial field studies, and the overall facies succession, is that the condensed diamict of LU3 was deposited when Ryder Glacier retreated far enough inland to become cut-off from the main fjord. In Sherard Osborn Fjord, a relatively deep, isolated marine embayment exists behind a prominent topographic high lying 40 km inland of the modern grounding zone (Fig. 11)."*

*To:*

*"The explanation that best fits evidence from terrestrial field studies, and the overall facies succession, is that the condensed diamict of LU3 was deposited when Ryder Glacier retreated inland. In Sherard Osborn Fjord, a relatively deep, isolated marine embayment exists behind a prominent topographic high lying 40 km inland of the modern grounding zone (Fig. 11)."*

20. Lines 515, 525, 565 suggest you refer to Detlef et al., in review, which provides important sea ice reconstructions and marine conditions for Petermann Fjord over the same time period.  See https://doi.org/10.5194/tc-2021-25

*We have not drawn comparisons with Detlef et al. (2021) that is also undergoing review at this time. There are a number of mutual co-authors on these papers so we are very aware of the work. Similar biomarker-based reconstructions of sea ice are being conducted on Lincoln Sea and Sherard Osborn Fjord sediments. We feel it is better to wait until these results are ready before a more detialed analysis of regional sea ice conditions is undertaken. Furthermore, we feel it is generally better to reserve citations to manuscripts that are accepted, and since these are both going through review at the same time, this is a bit tricky.*

---

## Author Comment (AC2)

**Responses to reviewer #2**

Reviewer 2 has pointed out some important details of the manuscript that need to be clarified in the revised version. They have also made a number of editorial/technical suggestions for improving the manuscript. We appreciate their input and first respond to the 5 'General' comments and then the 41 'Technical' comments that were made.

Included at the end of the response letter are the 3 new figures that we propose adding to the supplementary information of the revised manuscript.

General comments:
1. It doesn't really make sense to use uncalibrated 14C ages in the Geological Setting. I suggest that the existing radiocarbon dates from Kelly and Bennike (1992) should be re-calibrated using Marine20 and the same deltaR as the new marine cores.

> *On one hand we agree that it would be helpful to provide re-calibrated dates for the original data presented in Kelly and Bennike (1992). We can do this using the very large dR uncertainty we have adopted in this manuscript, making the published dates directly comparable to our result - these can be included in Table 1. However, in the Introduction/Background we prefer to continue using the raw 14C ages for their samples and refer the reader to Table 1 to see the equivalent calibrated date using Marine20 and our dR uncertainty envelope. Using the raw 14C ages in the main text ensures that the Introduction/background does not become outdated when new constraints on dR for the region emerge. In the Discussion we can report the calibrated age ranges for easier comparison with our data (when needed). Our dR of 300 +/- 300 is designed to provide a robust estimate that covers the ranges in published literature and is useful for dating the lithostratigraphic boundaries in our records.*

2) The result section is a mixture of descriptions and interpretations. Example line: 287-288, 299-300, 308-309, 317, 324, 336-338. I suggest to clearly divide the result section into two separate sub-sections: description followed by interpretation. This will allow the reader to assess the data and follow the logic in the interpretations.

> *We can re-organize this section of the results, ending each of the Lithologic Unit descriptions with some of the basic interpretations of the depositional environment. The broader interpretation of the facies succession will remain at the start of the Discussion.*

3) Figure 5 offers a great summary of the most important data. It would be really nice to compliment the figure with the CT scans from the suppl. material or the high-resolution picture from the XRF scanner. It is really a pity that the CT scans are hidden in the suppl. material.

> *We agree, and seeing the citation metrics for the article, it is clear that the images in the supplementary material are not being widely viewed. We will introduce a new figure in the manuscript that show the CT-image, lithologic units and location of radiocarbon dates for each core.*

4) The way the age of the individual units has been constrained differ from most studies as it uses the min. and max. ages from each unit to define the age range. However, as the radiocarbon dates are not always placed optimally at the boundaries between units this makes it difficult to compare the age ranges of the units between the different cores. I

suggest that an age-depth model for each sediment core is produced. This would make it possible to determine the age at the boundaries (with an uncertainty) and also allow for a figure to be made where the proxy data (from figure 5) is plotted on an age scale. This is standard procedure and it would make a great supplement to the discussion section 5.2-5.4.

*Although we present a large number of new 14C dates, we do not have the ability to generate meaningful, continuous age models for each of the cores. The vast majority of samples we investigated did not contain enough foraminifera to obtain radiocarbon dates, so we have not been able to date each core at the resolution, or exact depth, that would we want. At the same time, it is only LU5/4 and LU1b where we do not have dated samples within a few centimeters of the lithologic boundary.*

*We expect improved age models to emerge from future work. This could come from 1) additional radiocarbon dates, 2) Improved constraints on the local reservoir effect (i.e paired samples, 210Pb, 137Cs, tephra), and 3) Improved stratigraphic correlation between cores, and potentially independent age control, from generating paleosecular variation records – as was done by Reilly et al., 2019 for Petermann Glacier. These measurements are in progress, and may allow us to stack more dates from the different cores onto a master chronology. Therefore, we feel that while using this really broad dR, and in light of new data that should soon emerge, it is not the right time to start generating detailed, conventional age models for each core.*

*We strongly feel that the approach we have taken here, is a robust way to date the major events and environmental changes that occurred through the Holocene.* ***Because we have applied such a large dR, improved constraints on dR and/or applying a baysian modeling technique, will narrow the uncertainty ranges of the unit boundaries. Furthermore, since we present our data as maximum constraints for the onset of the different units (using the youngest date from the underlying unit), they will remain correct.***

*However, we recognize that presenting more conventional age-depth models is somewhat necessary to evaluate some of the individual ages, and elucidate the reasoning behind 'accepting' or 'rejecting' some of the returned dates. In particular, the reviewer has later asked questions regarding why we rejected some dates from 10-GC and 8-PC. As suggested, we have compiled age-depth figures for each core (Review Fig. 1), and can include this in the supplementary information. While these are not age models per se, they do illustrate the stratigraphic ordering of dates in each core, and how they align with our proposed unit boundary ages.*

*In doing this, there are two important insights we have drawn that we realize must be explained better in the revised manuscript.*

*The first is to explain why the young age returned from the LU3/LU2 boundary in 10GC is not used to date this boundary (comment 21 by Reviewer 2 below). The reasons for not using this date for the base of LU2 are 1) older ages that are in stratigraphic order are found in the other cores for LU2, 2) There is strong evidence for a period of erosion/non-deposition across the boundary at 10GC on the outer sill (Review Fig. 2 to be included in supplementary material), and 3) This sample in 10GC comprised 4 cm of material (50-54 cm), with some of this coming from the overlying LU2 and some from below in LU3. It is not possible to identify how many of the dated specimens came from each unit.*

*The second point we will more fully illustrate and discuss is why we identified the lowest date in 8-PC (near the base of LU5) as an outlier. There are two reasons for this that we will highlight in the revised manuscript. 1) The age difference between the lowest two dates in 8-PC is 1680 years, but they are only separated by 24.5 cm, suggesting that the lowest most age is too old, or deposition was not continuous, and 2) the older, lowest most age was obtained from a sample containing mixed benthic foraminfera, which in general seemed to be more prone to returning old ages compared to mono-specific samples of C. neoteretis. More clearly recognizing that sedimentation during LU5 was not continuous is important. The laminated unit clearly contains numerous erosional zones of unknown duration, which are logically more frequent near the base of the unit, in what we interpret as a grounding zone proximal setting. We have added symbols indicating this on our compilation of 14C dates (Figure 8 in original manuscript), and include an additional detailed interpreted image containing examples from 7-PC, 8-PC and 9-PC to the supplementary material (Review Figure 3).*

5) It is clear that the most challenging unit to interpret is the diamicton unit 3. The unit differs from most other units which are laminated. It only resembles unit 6 the lowermost unit which is interpreted as subglacial till. However, the authors prefer an alternative explanation where unit 3 represents massive IRD deposition during a period where the ice front is most retracted. They also discuss other possibilities but find them less likely. I am not completely convinced but agree that it is difficult the interpretation of unit 3 is not straight forward. I wonder if unit 4 instead could represent the period where RG is most retracted and that unit 3 represents the phase where it begins to readvance sending icebergs (IRD) into Sherard Osborn Fjord again. If correct, the onset of readvance is c. 6 cal ka BP which coincides with the general cooling trend in the Agassiz ice core record.

*We should start by clarifying that we do not argue for 'massive IRD deposition' during LU3 – which implies high rates of IRD input – but rather slow and sustained deposition of IRD in the absence of significant meltwater derived sediments that contributed to deposition of LU5, 4, 2 and 1. We think this is clear in the manuscript and wanted to clarify this here, as we are unsure if the reviewer meant 'rapidly deposited' when they say 'massive IRD deposition'.*

*Is it possible that LU4 represents the most retracted phase of Ryder Glacier, and LU3 an advance? We do not believe so. One basic reason is the overall facies succession. If LU3 represented a re-advance, it becomes hard to understand how this can transition into potentially seasonally open water conditions, and generally ameliorated climate conditions, during the deposition of LU2. Our interpretation is also consistent with what has been described by Kelly and Bennike (1992) based on mapping and dating of raised shorelines and morraines. In particular, as we state in the **Geologic, oceanographic and glaciologic setting**: ". . . peat deposits over which the ice margin advanced provide an age of 5100±130 14C a BP (Station 41), while at Steensby Glacier, reworked marine macrofossils in lateral moraines yield an age of 4870±80 14C a BP (Station 34; Kelly and Bennike, 1992).". Calibrated using Intcal20 and Marine20 respectively, these provide ages of 5830 ± 170 cal a BP (for the peat) and 4560 ± 410 cal a BP (for the reworked molluscs on top of the lateral morraines). These are more consistent with our interpretation that the re-advance occurred during LU3/LU2. Obviously this requires some clarification in the **Discussion** of the manuscript where we will more closely tie the earlier findings of Kelly and Bennike*

*(1992) to our results. Providing updated calibrated ages in Table 1 will help with this comparison. **Importantly, in one sense the reviewer is correct, we do not know when the re-advance began. In the revised manuscript, we will point out that the initial re-advance likely began during LU3, but critically, a marine based glacier and ice tongue were definitely established by the onset of LU2.** This remains consistent with our interpretation of the facies succession.*

Technical comments:
1. Line 20: Change to Greenland Ice Sheet.
> *This has been corrected.*

2. Figure 1: Add Gl. for glacier after Humboldt, Petermann etc. Also add Ice Sheet after Greenland.
> *This has been corrected.*

3. Line 43: Change to Greenland Ice Sheet.
> *This has been corrected.*

4. Line 61: Change to Last Glacial Maximum.
> *This has been corrected.*

5. Line 66: Change to Möller.
> *This has been corrected in the main text and reference list.*

6. Line 75: Add glaciers after Petermann.
> *This has been added.*

7. Line 75: Change to Nioghalvfjerdsfjord Glacier.
> *This has been added.*

8. Line 110: Change to north Greenland.
> *This has been corrected.*

9. Line 118: 9390+-90 date is not in table 1
> *This was an oversight and the date has now been included in the table.*

10. Table 1: Combine with Table and calibrate the old ages with Marine20.
> *We can include re-calibrated Marine20 dates in the table using the same dR as we apply to make comparisons more straightforward.*

11. Line 127: Delete cal a BP after >9.5.
> *This has been corrected.*

12. Line 129: Mark the ice-dammed lake on the map.
> *To our knowledge the limits and extent of the proposed ice dammed lake have not been mapped out. For this reason, we have not portrayed its extent on the figure. It is not clear whether it would have extended from Sherard Osborn Fjord across to Victoria Fjord, or occupied a more restricted part of Wulff land – which would ultimately depend on the glacier(s) configuration and relative sea level at the time of its existence.*

13. Line 131: Are the dated shells reworked into the moraine?

*Good question. In fact we had made a slight error in reporting the station names and mixed up station 36 and 40. The problem is that a clear station description is not available for some instances, or hard to interpret from the summary work be Kelly and Bennike, 1992. We have gone back to the original publications describing the dated material (Kelly and Bennike, 1985, Bennike and Kelly, 1987), and these also do not provide a very good geological context for the samples. We have added more detailed sample local descriptions in Table 1, and corrected our previous error. In short, at station 36, in western Warming Land, the dated shells were from marine silts that were younger than the Warming Land Stade morraines and returned and age of 8210±120 $^{14}$C a BP (not 6480±100 $^{14}$C a BP that we originally wrote). The age of 6480±100 $^{14}$C a BP came from shells from a marine silt in front of Ryder Glacier (Station 40) and from what we understand, constrains the timing for ice retreat towards the modern position.*

*Bennike, O. & Kelly, M. 1987: Radiocarbon dating of samples collected during the 1984 expedition to North Greenland. Rapp. Grønlands geol. Unders. 135, 8-10.*

*Kelly, M. & Bennike, O. 1985: Quaternary geology of parts of central and western North Greenland: a preliminary account. Rapp. Grønlands geol. Unders. 126, 111-116.*

14. Line 136: Change to Ryder Glacier.

*This has been corrected.*

15. Table 2: Could be moved to suppl. material.

*Table 2 contains the meta-data for the coring stations. We prefer to leave the table in the main paper, as it is an important resource for future studies. Those who want to locate the cores geographically etc, should not need to dig through the supplementary material.*

16. Figure 4: Is not showing much and could be moved to suppl. material.

*Once again, we prefer to keep this in the main manuscript. We agree that a more detailed assessment of the subbottom data should be undertaken in future studies. However, in the context of this manuscript, this figure clearly illustrates the general thickness of sediments on top of acoustic basement, and how far the cores penetrated into this sedimentary cover.*

17. Line 167-168: Change lithified to compacted.

*This was also suggested by Reviewer 1, and has been changed to 'consolidated'.*

18. Table 3: 13C is missing for sample 26.

*We have added 'N/A' to the table, as the sample was too small to provide a 13C measurement.*

19. Line 268: Delete glacial after Holocene.

*This has been deleted.*

20. Line 271: Change through to and.

*This has been changed.*

21. Figure 5:
On 10-GC the 2450 date seems to be within unit 3 but it is marked as unit 2 in figure 8?

*Please see our detailed response to General comment 4. Essentially, this date was obtained from a 4-cm thick sample that included sediments from LU3 and LU2, and we believe thata hiatus exists between these units at this specific station.*

On 7-PC the date 7090 seems to be an outlier but it is not marked with red.

*Two reasons for this: 1) The two ages at this depth overlap at 1-sigma. Obvisouly they will not if the dR is reduced, but for now this was a basic criteria we applied to identify outliers. 2) The two dates from that interval are from a planktic and a benthic sample, and we cannot be sure that different dR values would not resolve the apparent offset. Therefore, we have not discarded either date at this time.*

Also, what is the square next to 7090 representing?

*This was from an earlier version of the figure and used to differentiate whether it was from a planktic or benthic foram sample. We did not mean to carry this convention over to the published manuscript, and have made all sample locations circles.*

Why is the last date in core 8-PC/GC an outlier?
*Please see our detailed response to General comment 4. We have adjusted our interpretation and now use this age to date the base of LU5, and provide evidence that sedimentation on the inner sill during LU5 was discontinuous.*

22. Figure 8: I don't understand why the 14C dates in this plot have a normal distribution? Also see general comment 3.

*In this figure we have shown the likelihood distribution, mean and 1-sigma range for each sample when calibrated using Marine20 and a dR of 300+/-300 years. They are not all exactly normally distributed, but they do no have the skewness that would arise if we were modeling the ages and showed a posterior distribution. We can clarify this in the figure caption.*

23. Table 4: Not important and can be omitted if the age depth models as suggested in general comment 4 will be made.

*In line with our response to general comment 4, we provide these conventional 'age-depth' models in the supplementary material. However, we strongly believe that the approach of dating the lithoistratigraphic boundaries makes most sense at this time. Any conventional age models we publish at this point will most certainly be re-vised in the near future. Although this will reduce the uncertainty in the ages for the LU's they will very likely to stay within the reported age range we have defined because of the very large dR we have used.*

24. Line 393: Change to Northern Hemisphere.
*This has been changed.*

25. Figure 9: Really nice illustration – are the radiocarbon ages from Ellesmere re-calibrated?
*Yes, they were recalibrated using Intcal20. We should have indicated this in the figure caption and will do so in the revised ms.*

26. Line 419: Change to Möller.
*This has been changed.*

27. Line 423: Add cal ka BP after 12.5.
*This has been changed.*

28. Line 430: Delete one l in Fullford.
*This has been corrected.*

29. Line 435: Change to GrIS.
*This has been changed.*

30. Figure 10. Again, a really great illustration. Could you add the locations of the Warming Land and Kap Fulford Stades on the figure?
*These have been added to the figure.*

31. Line 466: It is stated that …LU3 range from 6.3 to 3.9 cal ka BP. However, the upper part of unit 3 in 10-GC is 2450 14C a BP. Why is this date omitted in the summary?
*Please see our detailed response to General comment 4.*

32. Line 508-510: Temperatures were not 2.5-4ºC warmer until 6.2-6 ka. They were still high but the peak warmth occurred in the beginning of the Early Holocene and was insolation driven.
*We have adjusted this sentence to read:  "and peak late summer air temperatures inferred from δ18O of chironomids in Secret and Deltasø lakes that were >2oC warmer then present until 6.2 cal ka BP (Axford et al., 2019; Lasher et al., 2017)" as this is what they report in their work (see Fig. 6 in Axford et al., 2019).*

33. Line 517: Zekollari models suggest that at least part of the Hans Tausen ice cap survived the HTM.
*We were aware of this, and had stated that the southern dome of the ice cap had disappeared. However, we can make this more clear by statin::*
*"This timing for glacier advance is consistent with cooling seen in lake based temperature reconstructions around 4 cal ka BP (Lasher et al., 2017) and the oldest estimated age (3.5 to 4.0 cal ka BP) for ice at the base of the southern dome of Hans Tausen ice cap, which had disappeared during the Middle Holocene - although northern parts of the ice cap had survived (Madsen and Thorsteinsson, 2001; Landvik et al., 2001; Zekollari et al., 2017)."*

34. Line 520: Change to: Middle Holocene.
*This has been changed.*

35. Line 521: Change to: GrIS.
*This has been changed.*

Figure 11: Again, a great illustration. Maybe consider changing the white color of the modern ice limit to red. It would also be great to get the Kap Fuldford and Warming Land moraines on the maps.
*These have been added to the figure.*

36. Line 540:  or 2450 14C a BP? See comment Line 466.

*We are now clear in the manuscript that this age is not used to date the base of LU2, because we interpret that the transition from LU3 to LU2 at 10-GC is non-conformable.*

37. Line 564: Change to Funder et al., 2011.
   ***This has been corrected.***

38. Line 585: north Greenland.
   ***This has been corrected.***

39. Line 600: Søndergaard et al (2020) have published a paper in Climate of the Past on the deglaciation on Inglefield Land, Smith Sound and nares Strait that would fit into the discussion.
   *We have added this reference to section 5.3 in the Discussion (5.3 Middle Holocene inland retreat and collapse of Ryder's ice tongue)*

40. Line 601: Can the differences in fjord physiography play a role in the different timing of retreat between Petermann and Ryder glaciers? Sherard Osborn Fjord is deeper and potentially more susceptible to dynamic ice retreat compared to the shallower Petermann fjord.
   *Just below this we argue (lines 610-625) that the physiography of Sherard Osborn fjord is conducive to glacier and ice shelf stability. We state that there is little evidence for collapse or surge events, in the form of IRD pulses. Given the arguments we have already laid out, we do not feel the need to backtrack and say the physiography of the fjord could have been a key factor in Ryder's retreat from the coast.*

41. Line 650: Change to: Late Holocene.
   ***This has been corrected.***

[Figure]

hiatus or erosion

● ¹⁴C dated samples

■ Derived minimum ages for the base of lithologic units

Review Figure 1: Age-Depth plots of the calibrated 14C dates (circles) and derived ages for the LU boundaries (squares)

[Figure]

Review Figure 2: Transition from LU3 to LU2 in 10-GC. Here there is an aburpt transition that, unlike the other stations is not biotubated. We interpret this as being an eorsional surface. The position of 10-GC-1, 50-54 cm is shown. This sample was not used to date the base of LU2 because it straddled the LU boundary and crossed what we believe is an erosional surface.

¹⁴C sample range

[Figure]

Review Figure 3. Interpreted evidence for erosion during the deposition of LU5 based on CT-scanning data. At 7-PC truncated laminae appear frequently near the base of LU5. In 8-PC and 9-PC, from the inner sill, LU5 was much thinner and had evidence for possible erosional episodes throughout. In some instances it is unclear whether laminae are truncated by erosion or offset by erosion. Yellow triangles indicate positions of radiocarbon dates. The numbers correspond to sample ID's in Table 3 of the main text.

---

## Author Response (AR2)

**tc-2021-95: Technical Corrections**

We appreciate the careful review of our revised manuscript and acknowledge that we failed to properly align some parts of the revised manuscript with what we had written in the response letter. This was an oversight by me, and I have corrected these issues in the revised paper. In the tracked changes version, we have indicated the sections that have been revised by highlighting them in yellow.

In response to the specific points and suggestions:

1. Reviewer 1, lines 286-288 on the tracked changes version do not quite match your response. *We have corrected this and now add (lines 289-290) that 'Although we recognize that deformed diamictons can be deposited beneath grounded ice (deformation till) or proximal to the grounding line (gravity flow deposits), both options imply deposition close to the grounding zone."*

2. Reviewer 1, line 507 on the tracked changes version (likewise). *We have added the corrected sentence from our response letter (line 516-518) "The explanation that best fits evidence from terrestrial field studies, and the overall facies succession, is that the condensed diamict of LU3 was deposited when Ryder Glacier retreated inland. In Sherard Osborn Fjord, a relatively deep, isolated marine embayment exists behind a prominent topographic high lying 40 km inland of the modern grounding zone (Fig. 12)."*

3. Reviewer 2, where you indicate that you will "more closely tie the earlier findings of Kelly and Bennike (1992) to our results.". I could not see where you did this in the Discussion with changes tracked. *This comment and our original response largely concerned the interpretation of LU3. Some support for our interpretation that it was a deposited while the ice margin was inland comes from terrestrial mapping and dating which we alluded to in the first submission but were not specific about the details. By providing calibrated ages (using a similar dR as applied to our samples) we had the opportunity to clarify this in the revised manuscript – but failed to do so. Now we have added a final paragraph to section 5.3 that addresses this. It reads (Lines 565 to 570) "In summary, our interpretation is that the end of LU3 (3.9 ± 0.4 cal a BP, Table 4) marks the re-growth of a marine based glacier and ice tongue. Importantly, this is consistent with existing dates constraining the onset of the Steensby Stade as described by Kelly and Bennike (1992). In particular, peat deposits over which Ryder's ice margin advanced during the Steensby Stade provide a Middle Holocene age of 5830 ± 170 cal a BP (Station 41), while reworked marine macrofossils in lateral moraines at Steensby Glacier yield an age of 4560 ± 410 cal a BP (Station 34; Kelly and Bennike, 1992). Therefore, while the re-advance of the local ice margin likely occurred prior to the Late Holocene, we argue that a marine-based glacier and ice tongue were not established until close to the Middle to Late Holocene transition."*

4. My only other (optional) comment would be that you might want to consider referring to Supplementary Figures more specifically in the revised manuscript, i.e. pointing the reader to the Supplementary Figure S7, rather than just 'Supplementary materials' more generally. *Excellent point and a major oversight on our part. We now ensure that all the supplementary figures are specifically referred to in the text, at the appropriate time.*

*While correcting this, we also realized that some additional information had to be added to the main text, and minor adjustments made to Figure 9 and Figure S7. We have included a new paragraph in section 4.1 that discusses the evidence for erosion in Lu5 and a hiatus/erosion between LU3/LU2 in 10-GC. Examples of these are provided in the supplementary figures which were not called out in the previous version. The new paragraph (lines 414-419) reads:*

*"The transition between LU3 and LU2 in 10-GC is more abrupt than in other cores and is not bioturbated (Supplementary Figure S8). We infer a hiatus across this transition and do not use the younger age (Sample #41, Table 3) to date this boundary. Instead we rely on the numerous other older dates obtained from the base of LU2 to date this boundary (Fig. 9). Similarly, based on the occurrence of truncated laminae seen clearly in the CT-images, frequent intervals of erosion are indicated during deposition of LU5 (Fig. 9). Examples of this in 7-PC, 8-PC and 9-PC are provided in supplementary Figure S9."*

*The small changes to Figure 9 and Figure S7 amount to the insertion of a symbols indicating possible erosion in LU5 of 7-PC – making these figures consistent with the evidence we present in Figure S9.*

Sincerely,
Matt O'Regan and co-authors